# TileQ: Efficient Low-Rank Quantization of Mixture-of-Experts with 2D Tiling

**Hongyaoxing Gu** [1 2]  **Xinzhe Chen** [1 2]  **Lijuan Hu** [1]  **Liu Fangfang** [1 3]

## Abstract

Mixture-of-Experts (MoE) models achieve remarkable performance by sparsely activating specialized experts, yet their massive parameters in experts pose significant challenges for deployment. While low-rank quantization offers a promising route to compress MoE models, existing methods still incur nonnegligible memory overhead and inference latency. To address these limitations, we propose TILEQ, a fine-tuning-free post-training quantization (PTQ) method that employs 2D-tiling structured low-rank quantization to share low-rank factors across both input and output dimensions of MoE experts. Furthermore, we introduce an efficient inference technique for TILEQ that fuses multiple low-rank expert computations into a single-pass operation, significantly improving hardware utilization. Experiments show that TILEQ cuts down additional memory usage up to $10\times$ and reduces inference latency to $\sim 5\%$ while preserving state-of-the-art accuracy. Our code is available at: https://github.com/grysgreat/TileQ

## 1. Introduction

In recent years, the MoE architecture has emerged as a powerful paradigm for scaling large language models (LLMs), achieving remarkable success (Guo et al., 2025). MoE models distribute computational workloads across a set of "experts," and employ sparse activation mechanisms such that only a small subset of experts is activated during each inference step. This design enables MoE models to deliver performance comparable to larger dense models while substantially reducing computational costs (Muennighoff et al., 2025; Lin et al., 2024a; Kim et al., 2023).

However, despite their computational efficiency, MoE architectures introduce a critical challenge: substantial memory overhead. Although each input token activates only a small subset of experts during inference, the entire set of model parameters must remain resident in memory (Rajbhandari et al., 2022). This large memory footprint severely limits the deployment of MoE models on resource-constrained hardware and significantly increases the cost of cloud inference.

To address the memory bottleneck, model compression techniques such as quantization have become a crucial avenue of research (Li et al., 2025a; Frantar et al., 2023; Li et al., 2026). Consequently, developing advanced MoE quantization methods is essential for lowering deployment barriers and inference costs. Among existing approaches, PTQ (Frantar et al., 2023; Hubara et al., 2021; Chen et al., 2025c; Zhang et al., 2025) offers significant practical advantages over quantization-aware training (QAT) (Chen et al., 2025a), which enabling rapid deployment with minimal computational overhead without retraining. Moreover, recent SOTA PTQs have narrowed the accuracy gap with QAT (Tseng et al., 2024b; Lin et al., 2024b), making it particularly well-suited for MoE models that prioritize efficient inference without compromising fidelity.

Among PTQs, low-rank methods have demonstrated strong performance in MoE quantization (Huang et al., 2025; Li et al., 2025b). **Nevertheless, the challenge lies in the trade-off among three key points**: **(i).**Simple quantization schemes suffer from severe accuracy degradation under extremely low bit-widths; **(ii).**Experts are small and sparse in the MoE models, where low-rank PTQs often incur significant additional memory overhead; **(iii).**High-accuracy quantization methods typically introduce considerable inference latency. Moreover, fine-tuning (FT) methods degrade model generalization, limiting their applicability for rapid task adaptation (Zhang et al., 2024b;a).

To tackle these challenges, our aim is *'achieving high accuracy and compression ratios while keeping minimal inference latency'*. Our contributions are as follows:

- To overcome the limitation of traditional low-rank methods, we propose TILEQ—a fine-tuning-free PTQ approach for MoE models that leverages a 2D-tiling layout based on singular subspace clustering to enable efficient shared representations across experts.

[1]Institute of Software Chinese Academy of Sciences [2]University of Chinese Academy of Sciences [3]Key Laboratory of System Software (Chinese Academy of Sciences). Correspondence to: Liu Fangfang <fangfang@iscas.ac.cn>.

*Proceedings of the $43^{rd}$ International Conference on Machine Learning*, Seoul, South Korea. PMLR 306, 2026. Copyright 2026 by the author(s).

- We proposed an optimized fusion algorithm that combines multiple low-rank expert GPU kernels across all tokens into a single-pass operation, significantly accelerating both the prefill and decode phases.
- Experiments show that TILEQ executes **swiftly** without **fine-tuning** and achieves **state-of-the-art** quantization accuracy while saving up to **90%** low-rank extra memory and reducing inference latency to **5%**.

## 2. Related Works

### 2.1. Post-Training-Quantization

In post-training quantization (PTQ) for large language models, existing methodologies can be systematically organized into three hierarchical categories:

**Base Format.** These foundational quantization schemes establish the core framework for PTQ:

- *Symmetric Format* (Nagel et al., 2020): Employs a symmetric grid centered at zero to simplify arithmetic operations, but is highly susceptible to outliers.
- *Asymmetric Format*: Introduces new formats or learnable zero-point offset (Li et al., 2025d; Rusci et al., 2020) to better align the quantization range with the empirical weight distribution.
- *Vector Format* (Chee et al., 2023; Van Baalen et al., 2024; Liu et al., 2024): Represents groups of weights as vectors and compresses them by clustering or codebook-based encoding.

**Optimization methods.** These techniques build upon base methods by incorporating model-aware refinements to mitigate quantization-induced degradation:

- *Hessian optimization* (Frantar et al., 2023): Optimizes quantized weights $Q$ by minimizing a Hessian-weighted reconstruction loss, typically formulated as,

$$\mathcal{L}(W) = E_X \left[ \| \boldsymbol{W} \boldsymbol{X} - \hat{\boldsymbol{W}} \boldsymbol{X} \| \right], \qquad (1)$$

  where $H$ is a proxy Hessian matrix.
- *Activation-aware scaling*: Identifies channels with salient activation magnitudes and adjusts their corresponding weight scales to minimize downstream performance loss (Lin et al., 2024c; Xiao et al., 2023; Zhao et al., 2026a).
- *Outlier mitigation*: Addresses outliers through:
  - *Rotation*: Exploits the orthogonality invariance through Hadamard rotations (Ashkboos et al., 2024; Chee et al., 2023; Lin et al., 2024b) or learnable orthogonal matrices (Liu et al., 2025).
  - *Clipping*: Cutoff outliers to constrain numerical range, thereby stabilizing quantization and improving overall fidelity (Shao et al., 2023).

- *Low-rank and sparsification*: Decomposes weight matrices into low-rank (Li et al., 2025b; Gu et al., 2026; Zhao et al., 2026b) factors or enforces structured sparsity (Dettmers et al., 2024), lowering both memory footprint and error propagation during quantization.

**Fine-tuning.** These approaches recover lost accuracy through lightweight adaptation, though task-specific fine-tuning may compromise model generality:

- *LoRA-based FT* (Dettmers et al., 2023; Zhang et al., 2024b): Integrates low-rank adapters into the quantized model and updates only these lightweight modules, achieving significant performance recovery with minimal overhead.
- *Self-supervised FT*: Leverages unlabeled data and self-supervised objectives to recalibrate internal representations of the quantized model (Chen et al., 2020).

Contemporary PTQs typically compose these strategies in a modular fashion: they begin with a base format, enhance it with one or more optimization techniques tailored to model architecture and data characteristics, and optionally apply FT when higher accuracy is required.

### 2.2. Low-Rank Methods in MoE Quantization

In MoE models, about 95% of the parameters reside in the MLP experts and low-rank methods has emerged as a promising direction for MoE quantization due to its ability to simultaneously reduce the storage consumption of experts. Formally, low-rank quantization approximates a weight matrix $\boldsymbol{W} \in \mathbb{R}^{i \times o}$ as:

$$\boldsymbol{W} \approx Q(\boldsymbol{W} - \boldsymbol{W}_r) + \boldsymbol{W}_r \,;\, \boldsymbol{W}_r = \boldsymbol{U}\boldsymbol{V} = \text{SVD}(\boldsymbol{W}), \quad (2)$$

where $\boldsymbol{U} \in \mathbb{R}^{i \times r}$, $\boldsymbol{V} \in \mathbb{R}^{r \times o}$ and $r \ll \min(i, o)$ stands for the rank and $Q(\cdot)$ denotes a quantization operator applied to the low-rank factors. Traditional methods applies low-rank quantization to each expert independently, without considering relationships or redundancies across other experts and typically stored in 16-bit (Zhang et al., 2024a; Li et al., 2025b; Yang et al., 2024; Chen et al., 2025b). However, these methods yields limited compression gains when the number of experts $K$ is large.

To address this limitation, PTQs for MoE apply shared low-rank basis $\boldsymbol{U} \in \mathbb{R}^{i \times r}$ and specific $\boldsymbol{V} \in \mathbb{R}^{r \times (o \times K)}$ in 1D-tiling to capture common structures across all experts (Li et al., 2025c).

Although the shared singular method achieves a higher compression ratio, it only cutoff half the memory costs and still introduces significant memory overhead in sparse MoE models (Team, 2025). Furthermore, during inference, the $\boldsymbol{V}$ matrices of different experts remain disjoint, necessitating separate GEMM operations for each expert. The absence of

computational fusion further leads to an additional latency overhead of more than **50%** (Huang et al., 2025).

To address both the compression ceiling and inference inefficiency, we propose a **fine-tuning-free** PTQ method TILEQ, achieving significantly **higher compression** ratios and **minimum latency** in inference.

## 3. Method

**Model:** Denote that each MoE module has $K$ experts and includes a router $G$, so that $\mathcal{W} = \{W_k \in \mathbb{R}^{o \times i}\}_{k=1}^K$. For an input $X \in \mathbb{R}^{B \times i}$, the output is given by

$$\text{MoE}(X) = \sum_{W_k \in e_{k,X}} G_k(X) \cdot W_k(X), \qquad (3)$$

where $G(x)$ provides routing scores and $e_{k,x}$ denotes the set of selected top-$\mathcal{K}$ experts. The resulting output is a weighted combination of the outputs of the selected experts.

In this section, we focus on the experts in MoE models and propose a novel 2D low-rank PTQ method. Based on the proposed 2D-Tiling format (§3.1), we further introduce a fusion algorithm to enhance the inference speed for MoE models in both prefilling and decoding (§3.2).

### 3.1. TileQ: 2D-Tiling Structured Low-Rank Quantization for MoE Models

> **Motivation.** In MoE models, expert similarity serves as a key criterion for enabling efficient pruning and compression or redundant experts can be identified by the gating matrix (Li et al., 2025c; Frantar & Alistarh, 2023). However, existing low-rank methods either decompose each expert independently or concatenate all experts in one dimension to share singular $U$ or $V$, resulting in limited compression ratio and non-negligible inference latency (Huang et al., 2025).

**Solution.** Inspired by expert similarity, we propose a fine-tuning-free low-rank PTQ method as shown in Figure 1: all experts are arranged into a 2D tiled layout, explicitly leveraging similarity along both row and column to jointly share the $U$ and $V$ matrices in $(K \to M * N)$ — for an expert weight $W_k$ in Eq.3,

$$W_k = U_p \Sigma V_q + \hat{W}_k, \quad \{p, q\} = \phi_k \qquad (4)$$

where $U = [U_1, U_2, \cdots, U_M]^\top \in \mathbb{R}^{(M \times i) \times r}$ and $V = [V_1, V_2, \cdots, V_N] \in \mathbb{R}^{r \times (N \times o)}$; $\phi$ is a position table that maps the weights at the original $t$-th expert to $\{p, q\}$-blocks in the 2D low-rank matrix; $\hat{W}_k$ is a quantization matrix.

**Step 1: Scaling.** Since important activation values significantly affect quantization accuracy (Lin et al., 2024c), a diagonal scaling matrix is encoded into the weight matrix at the beginning. For each $W_k$, form

$$\tilde{W}_k^{scale} = W_k s_k \in \mathbb{R}^{i \times o}, \qquad (5)$$

where $s_k$ is a scaling vector calculated by calibration. This transposition ensures that the left singular vectors of $\tilde{W}_k$ correspond to the input feature space (facilitating clustering of $U$), while the right match the output (for $V$ clustering).

**Step 2: Extracting.** For each $\tilde{W}_k^{scale}$, we compute an approximate rank-$r_0$ approximation:

$$\tilde{W}_k^{scale} \approx U_k \Sigma_k V_k^\top, \qquad (6)$$

then vectorization flatten and normalize $U_k, V_k$ into $\mathbb{R}^{ri}$ and $\mathbb{R}^{ro}$ to extract norm feature embeddings:

$$u_k = \frac{\text{vec}(U_k)}{\| \text{vec}(U_k) \|_2}, \quad v_k = \frac{\text{vec}(V_k)}{\| \text{vec}(V_k) \|_2}. \qquad (7)$$

**Step 3: Clustering.** We perform biclustering on the normalized embeddings to assign each expert $k$ an ideal grid coordinate $(m_k, n_k)$. Then apply KMeans to obtain:

$$\begin{cases} \{\mathcal{G}_m^{\text{row}}\}_{m=0}^{M-1} = \text{KMeans}(\{\bar{u}_k\}_{k=1}^K, n_{clusters} = M), \\ \{\mathcal{G}_n^{\text{col}}\}_{n=0}^{N-1} = \text{KMeans}(\{\bar{v}_k\}_{k=1}^K, n_{clusters} = N), \end{cases} \qquad (8)$$

where $m$ and $n$ denote the cluster indices of row and column and the resulting partitions are defined as $\mathcal{G}_m^{\text{row}} = \{k \mid r_k = m\}$ and $\mathcal{G}_n^{\text{col}} = \{k \mid c_k = N\}$. This assignment encourages experts with similar singular matrix to share rows or columns, yielding the ideal tiling $(m_k, n_k)$ for expert $k$.

**Step 4: 2D-Tiling.** The ideal tiling in ¶**Step 3** is hardly met, as multiple experts may be simultaneously assigned to the same position. The mapping must satisfy: **(i)** no two blocks occupy the same tile; **(ii)** each block remains as close as possible to its ideal location.

Formally, we seek a placement function $\phi(k) \mapsto (m, n)$ that approximately solves

$$\min_\phi \sum_{k=1}^K \| \phi(k) - (m_k, n_k) \|_1. \qquad (9)$$

To solve this, we use a greedy spatial allocation strategy. For each position $(m, n)$, we define its anchor in the physical grid as $(m^\star, n^\star) = (\min(m, M-1), \min(n, N-1))$. Blocks are placed in unoccupied cells around the anchor using a concentric search ordered by Chebyshev distance:

$$\mathcal{N}_\rho = \{(m', n') \in \mathcal{P} \mid \|(m', n') - (m^\star, n^\star)\|_\infty = \rho\}, \qquad (10)$$

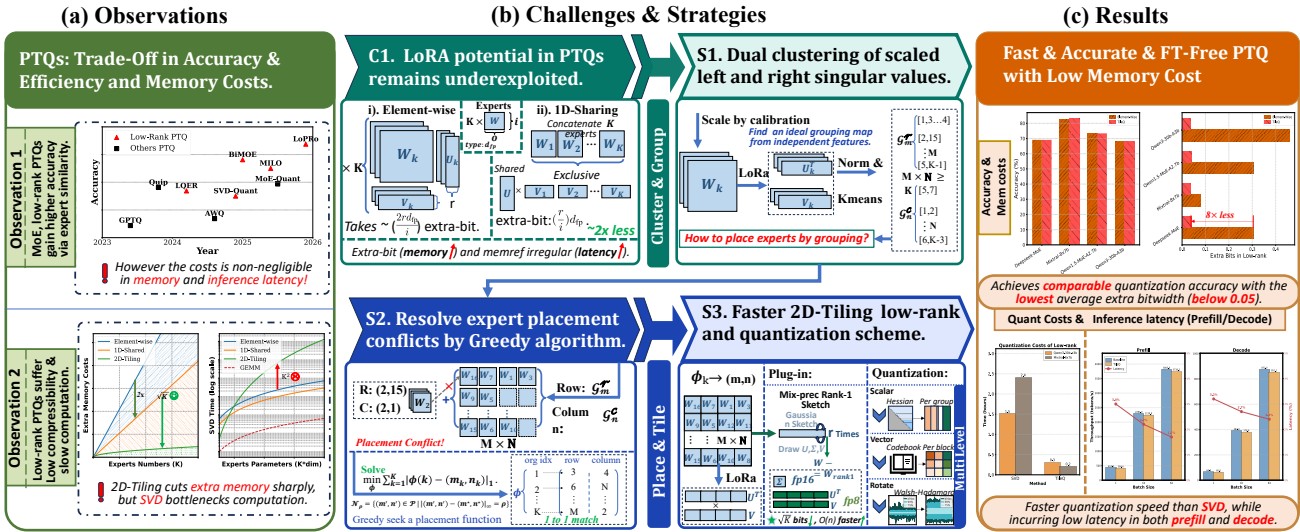

*Figure 1.* Overview of TILEQ. (a) discusses the observations and motivations of low-rank PTQs, (b) presents the challenges and strategies of our algorithm, and (c) shows the evaluations results. The pseudo-code is shown in Algorithm 1.

for $\rho = 0, 1, 2, \ldots$. Once placements $\{(m_k, n_k)\}_{k=1}^K$ are determined, we construct the tiling matrix $W_{\text{big}}$ as:

$$W_{\text{big}} = \sum_{k=1}^K \mathcal{E}_{m_k, n_k} \otimes \tilde{W}_k \in \mathbb{R}^{Mi \times No}, \quad (11)$$

where $\mathcal{E}_{m,n} \in \{0,1\}^{M \times N}$ is the basis matrix with 1 at $(m, n)$, and $\otimes$ denotes the Kronecker product in block (i.e., placing $\tilde{W}_k$ at the $(m_k, n_k)$-th tile).

The resulting mapping is $\phi(k) \mapsto (m_k, n_k)$, and the rank-$r$ shared tiling matrix is formed as:

$$\tilde{W}_k = (W_{\text{big}})_{\phi(k)}^{rank} s_k^{-1} = U_{m_k} \Sigma V_{n_k}^\top s_k^{-1}, \quad (12)$$

Here, we obtain the first form in Eq 4, which corresponds to the low-rank component of TileQ.

**Step 5: Quantization.** Now we quantize the residual after expert low-rank decomposition:

$$\hat{W} = \text{Quant}(W - \tilde{W}) = \mathcal{Q}(R).$$

TileQ targets the low-rank component in MoE quantization and remains compatible with mainstream quantization methods on the residual. Aiming for optimal accuracy, we choose three strategies from §2.1: ❶Apply GPTQ/GPTVQ's hessian optimization for scalar- and vector-level quantization, respectively. ❷Incorporate LOPRO's (Gu et al., 2026) low-rank rotation to maximize quantization accuracy.

*Plug-in.** In TILEQ, since feature extraction ($K \times SVD(\mathbb{R}^{o \times i})$) and decomposition of the 2D-tiling matrix ($SVD(\mathbb{R}^{(M \times o) \times (N \times i)})$) are computationally expensive, we replace it with Gaussian Sketch approximation (Gul et al.,

2026). Calculate the rank-1 matrix $W_1 = U_1 \Sigma_1 V_1$ by:

$$U = \frac{Q}{\|Q\|}, \Sigma = \frac{\|Q^\top W\|}{\|Q\|}, V = \frac{Q^\top W}{\|Q^\top W\|}, \quad (13)$$

where $S \in \mathbb{R}^i$ is a Gaussian sketch vector, $it$ is the iteration times and $Q = (WW^\top)^{it} W \Sigma$. Then, we set $W = W - W_1$ and do the same process for $r$ times to have any rank approximation. Since the singular value matrix $\Sigma$ is diagonal with low storage cost, by storing $U$ and $V$ in *fp8* while retaining $\Sigma$ in *fp16*, the storage overhead can be reduced by half. This iterative approach achieves nearly identical to SVD while being way faster in computation.

> **Insights ♀.** TILEQ jointly structures MoE experts into a 2D-tiling low-rank layout quantization without fine-tuning. This pipeline explicitly exploits expert similarity to enable shared $U$ and $V$ factors across tiles. Our method is highly flexible, supporting quantization methods from scalar to vector-wise and other optimization. Furthermore, we replace costly SVD with a fast sketch-low-rank approximation and adopts a mixed-precision storage strategy, which simultaneously reduces quantization cost and improves model compression ratio.

### 3.2. Efficient Fused Sparse Low-Rank Inference in 2D-Tiling

> **Motivation.** Traditional low-rank PTQs only takes minor latency overhead (under 10%) in dense models (Li et al., 2025b). However, in MoE models, experts are much smaller, making the overhead non-negligible (from **15%** to more than **75%**). Shared-$U$ approach like MILO (Huang et al., 2025) reduces this

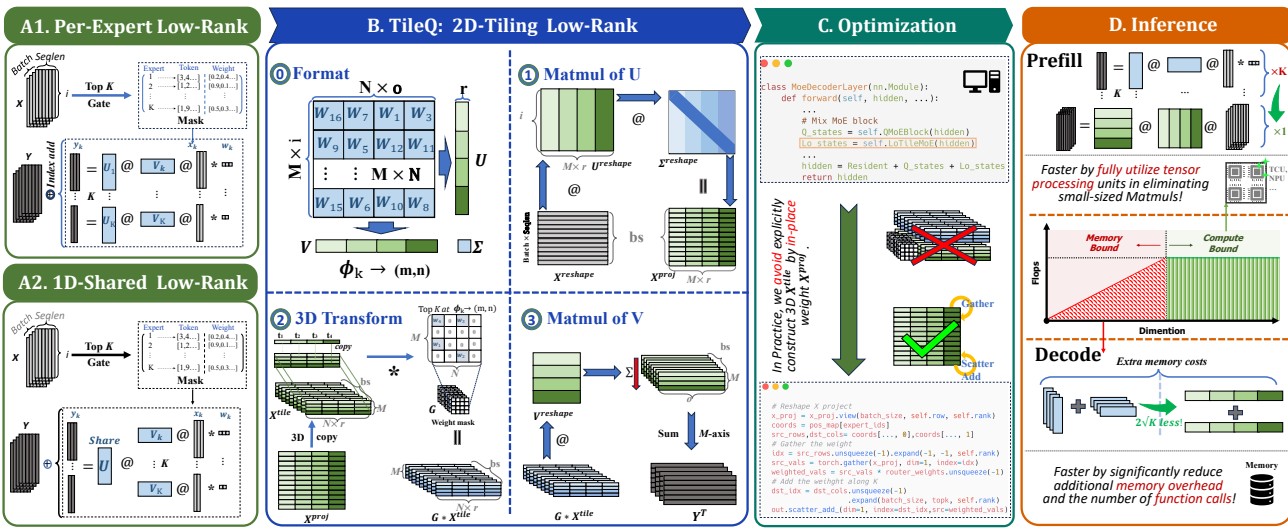

*Figure 2.* Efficient MoE with Low-Rank Decomposition in TILEQ: Enable 2D-Tiling to In-Place Optimization and Fast Inference

overhead to around **10%** on Mixtral-8x7B through kernel optimizations, but the latency rises to more than **50%** on **newer, sparser models** such as Qwen3-30B-A3B, significantly degrading practical usability.

**Solution.** We propose an efficient low-rank inference algorithm based on 2D-Tile low-rank shown in Figure 2, formulated as follows:

$$\text{MoE}(\boldsymbol{X}) = \text{QMoE}(\boldsymbol{X}) + \text{LoTileMoE}(X), \quad (14)$$

where QMoE denotes the standard quantized MoE forward pass, where the weights in Eq.3 are replaced with quantized matrices, ensuring compatibility with established inference frameworks (Kwon et al., 2023; Zheng et al., 2024). Meanwhile, LoTileMoE is an efficient inference method for the tiled low-rank. It exploits the low-rank structure to fused compute all low-rank experts and performs as follows:

For each token $b \in \{1, \ldots, B\}$ and its $t$-th $\in \{1, \ldots, \mathcal{K}\}$ selected expert. Denote the set is $E = \{e_{b,t} \mid b \in [B], t \in [\mathcal{K}]\}$ and $g_{b,t}$ is the routing weight from $\boldsymbol{G}$. The mapping $\phi$ assigns this expert to a unique tile in the 2D grid:

$$(m_{b,t}, n_{b,t}) := \phi(e_{b,t}),$$

Firstly, define the scaled left factor as $\boldsymbol{U\Sigma} \in \mathbb{R}^{(Mi) \times r}$ and project the entire batch:

$$\boldsymbol{X}_{\text{proj}} = \boldsymbol{X} \cdot (\boldsymbol{U\Sigma})_{\text{reshape}} \in \mathbb{R}^{B \times (Mr)},$$

where $(\boldsymbol{U\Sigma})_{\text{reshape}} \in \mathbb{R}^{i \times (Mr)}$ reshapes the tall matrix into feature-aligned blocks. This is then reshaped into a 3D tensor and broadcast across the column dimension:

$$\boldsymbol{X}_{\text{tile}} = \text{reshape}(\boldsymbol{X}_{\text{proj}}, [b, M, r]) \otimes \mathbf{1}_N^\top \in \mathbb{R}^{b \times M \times (rN)},$$

where $\otimes$ denotes channel-wise repetition. Next, we zero out all unselected tiles and inject routing-weighted contributions only at active expert locations. For each token–expert

pair $(b, t)$, the corresponding slice in $\boldsymbol{X}_{\text{tile}}$ spans columns $[n_{b,t}r, (n_{b,t}+1)r]$ within row $m_{b,t}$. We construct an output accumulator $\boldsymbol{A} \in \mathbb{R}^{b \times M \times (rN)}$, and update the accumulator as

$$\boldsymbol{A}[b, m_{b,t}]^{(n_{b,t})} \leftarrow g_{b,t} \cdot \boldsymbol{X}_{\text{tile}}[b, m_{b,t}]^{(n_{b,t})},$$

where the superscript $(n)$ denotes the $n$-th contiguous block of size $r$ along the last dimension (i.e., indices $[nr, (n+1)r)$). Then a batched matrix multiplication yields:

$$\boldsymbol{Y} = \boldsymbol{A} \cdot \boldsymbol{V}_{\text{reshape}}^\top \in \mathbb{R}^{b \times M \times o},$$

where the same reshape is performed to $\boldsymbol{V}$ as $\boldsymbol{U}$ and each row slice $\boldsymbol{Y}[b, m, :]$ corresponds to the contribution of all experts assigned to tile row $m$. The outputs are aggregated by summing over the top-$\mathcal{K}$ dimension:

$$\text{LoTileMoE}(\boldsymbol{X}) = \sum_{m=0}^{M-1} \boldsymbol{Y}[b, m, :] \in \mathbb{R}^{b \times o}. \quad (15)$$

This formulation replaces irregular sparse accesses with regular tensor operations (reshape, matrix multiplication $\cdots$), which are highly optimized on modern accelerators. The method thus achieves an optimal trade-off between memory efficiency and hardware utilization.

*****Tricks.** In implementation, we avoid explicitly constructing the accumulating tensors (Figure 2.C). Instead, we gather the routing-weighted activations directly into a compact 2D buffer $\boldsymbol{\Sigma} \in \mathbb{R}^{b \times (Nr)}$, where each column $n$ holds the sum of all activations assigned to that tile:

$$\boldsymbol{S}[b]^{(n)} = \sum_{t:\, n_{b,t}=n} g_{b,t} \, \boldsymbol{X}_{\text{proj}}[b]^{(m_{b,t})}, \quad \forall n \in [N]. \quad (16)$$

The final output is then computed as $\boldsymbol{S} \cdot \boldsymbol{V}_{\text{flat}}^\top$. This formulation reduces memory complexity from $O(bMrN)$ to

$O(bNr)$ and enables fully parallel execution without instantiating large intermediate tensors.

> **Insights 💡.** LoTileMoE enables efficient batched inference for low-rank quantization in 2D-tiling. It replaces irregular expert dispatch with regular tensor operations—projecting inputs once to eliminate dynamic loops and sparse memory accesses, thereby incurring **negligible latency** while leveraging highly optimized dense kernels (demonstrated in §4).

### 3.3. Analysis of TILEQ

**Compression ratio.** TILEQ achieves superior compression by exploiting 2D tiling across experts, achieving $\sqrt{K}$ lower storage cost than 1D baselines and $2\sqrt{K}$ less compared to element-wise methods. Especially beneficial in larger, sparser MoE models. (Detailed in Appendix A.3)

**Quantization Error.** In TILEQ, the error of $e^{(k)}$ satisfies

$$\mathcal{E}_{\text{TileQ}}^{(k)} \leq \mathcal{E}_{\text{ind}}^{(k)} + \epsilon_{cluster}, \tag{17}$$

where $\mathcal{E}_{\text{ind}}^{(k)}$ stands for the error of per-expert low-rank methods and $\epsilon_{cluster}$ is the error introduced by low-rank under 2D-Tiling. Consequently, TILEQ achieves quantization accuracy on par with independent per-expert compression. (Proved in Appendix A.2)

**Inference latency.** By replacing irregular per-token expert dispatch with dense GEMMs and vectorized operations, TILEQ avoids the kernel launch and reduces memory bandwidth bottleneck of MoE and achieves a complexity

$$C_{TileQ} = \mathcal{O}(BiMr + BNro + B\mathcal{K}r). \tag{18}$$

By aligning computation with tensor-specific hardware, TILEQ has significantly lower latency in both prefill and decode stages. (Detailed in Appendix A.4)

## 4. Evaluations

### 4.1. Setups

**Models:** Evaluations are carried out on a set of representative Mixture-of-Experts language models, including Qwen1.5-MoE-A2.7B (Team, 2024), Qwen3-30B-A3B, Qwen3-Next-80B-A3B (Team, 2025), Mixtral-8×7B (Jiang et al., 2024), and Deepseek-MoE-16B (Dai et al., 2024). Architectural details of each model are summarized in Table 2.

**Baselines:** We compare against GPTQ (standard per-channel PTQ) (qubitium, 2024), GPTVQ (vector quantization variant) (Van Baalen et al., 2024), LOPRO (low-rank rotation) (Gu et al., 2026) and MOEQUANT (Chen et al., 2025c) (MoE

quantization) in main evaluations with the same protocol; FP16 results serve as upper bounds.

**Metrics:** We evaluate model performance by multiple benchmarks: perplexity is measured on WikiText-2 (Merity et al., 2016), while down-stream tasks is assessed on: the AI2 Reasoning Challenge (Boratko et al., 2018), Physical Interaction Question Answering (Bisk et al., 2020), Winogrande (WI) (Sakaguchi et al., 2021), and the Massive Multitask Language Understanding benchmark (Hendrycks et al., 2020). All evaluations are conducted by `lm-eval` framework (Gao et al., 2024). Other experimental details are provided in Appendix B.

*Table 2.* Architectural configurations of experts in MoE models, with the number of experts reported as "regular + shared".

| MoE-Models | Experts | Top-K | Dimension |
|---|---|---|---|
| Qwen1.5-MoE-A2.7B | 60+4 | 4 | [1408,2048] |
| Qwen3-30B-A3B | 128+0 | 8 | [768,2048] |
| Qwen3-Next-80B-A3B | 512+1 | 10 | [512,2048] |
| Mixtral-8×7B | 8+0 | 2 | [14336,4096] |
| Deepseek-MoE-16B | 64+2 | 6 | [1408,2048] |

### 4.2. Main Results

We compare TILEQ with baselines on quantization performance, results are shown in Table 1 and can be analyzed from three directions. The comparison with other 2 MoE-Specific PTQ methods (MILO (Huang et al., 2025) and MXMOE (Duanmu et al., 2025)) are provided in Appendix D.2.

**Across bit-widths:** TILEQ demonstrates exceptional robustness under extreme low-bit settings, consistently outperforming all no-finetuning baselines—including GPTQ and GPTVQ. In the highly challenging 2-bit scenario, most methods suffer severe degradation (e.g., GPTQ on Mixtral-8x7B PPL drops from 3.87 to 15.3), whereas TILEQ$_v$ achieves a perplexity of 4.78; Under 3-bit quantization, performance further approaches FP16: TILEQ$_s$ attains a perplexity of 4.10 on Mixtral—nearly matching FP16 (3.87)—and scores 69.4 on MMLU, comparable to the unquantized model. This consistent cross-bit-width advantage stems from TILEQ's hybrid multi-layer quantization strategy. Moreover, compared to the element-wise method LoPRo, TILEQ achieves comparable or even higher accuracy, indicating that the fused expert features in MoE is effective.

**Quantization Strategies:** TILEQ offers both scalar (TILEQ$_s$) and vector (TILEQ$_v$) variants, revealing a fine-grained trade-off between accuracy and practical efficiency. At 2-bit, TILEQ$_v$ consistently achieves the best performance. This gain arises from its multidimensional codebook modeling intra-tile correlations. However, at 3-bit, the benefit of vector quantization diminishes: TILEQ$_s$ matches or slightly surpasses TILEQ$_v$ on key metrics, this stems from reduced

*Table 1.* Quantization results of TileQ and baseline methods under 2-bit and 3-bit settings. Here, Extra.Bits ($X_y$) denote the average bitwidths allocated to the group-wise quantization scales and the low-rank components, respectively. We report perplexity on WikiText-2 and accuracy on six downstream tasks: ARC-Challenge (AC), ARC-Easy (AE), PIQA (PQ), WinoGrande (WI), MMLU (MU), and HellaSwag (HS). MoEQ is in short of MOEQUANT. Further experimental details are provided in Appendix B.

| Models | Methods | Extra. Bits ↓ | 2-BiT Quantization | | | | | | | 3-BiT Quantization | | | | | | |
|---|---|---|---|---|---|---|---|---|---|---|---|---|---|---|---|---|
| | | | PPL ↓ | Accuracy ↑ | | | | | | PPL ↓ | Accuracy ↑ | | | | | |
| | | | Wiki | AC | AE | PQ | WI | MU | HS | Wiki | AC | AE | PQ | WI | MU | HS |
| DEEPSEEK-MOE-16B | FP16 | - | 6.11 | 45.1 | 75.9 | 78.6 | 70.5 | 45.1 | 59.7 | 6.11 | 45.1 | 75.9 | 78.6 | 70.5 | 45.1 | 59.7 |
| | GPTQ | $0.13_{0.0}$ | 24.1 | 24.4 | 50.2 | 66.5 | 53.3 | 25.3 | 45.3 | 7.17 | 32.3 | 66.7 | 76.1 | 67.9 | 37.3 | 53.7 |
| | GPTVQ | $0.13_{0.0}$ | 7.32 | 35.5 | 70.1 | 75.2 | 66.6 | 35.3 | 51.1 | 6.53 | 37.2 | 70.2 | 77.1 | 68.6 | 42.5 | 56.9 |
| | MOEQ | $0.0_0$ | 1e3 | 24.9 | 30.2 | 35.3 | 32.1 | 26.2 | 28.2 | 7.78 | 37.6 | 69.2 | 74.3 | 62.1 | 41.2 | 45.8 |
| | LOPRO | $0.43_{0.31}$ | 7.03 | 38.3 | 72.5 | 75.8 | 68.4 | 38.2 | 52.5 | **6.19** | 42.6 | 74.8 | 78.7 | 69.8 | **45.1** | 58.8 |
| | TILEQ$_s$ | **0.16**$_{0.04}$ | 7.06 | 38.1 | 72.7 | 76.1 | 68.5 | 37.9 | 52.2 | 6.24 | **43.2** | **75.0** | 78.6 | 69.9 | 44.3 | 58.3 |
| | TILEQ$_v$ | **0.16**$_{0.04}$ | **6.71** | **40.2** | **73.3** | **76.2** | **68.8** | **39.1** | **53.0** | 6.23 | **43.2** | 74.9 | **78.8** | **70.3** | 44.4 | **58.6** |
| MIXTRAL-8X7B | FP16 | - | 3.87 | 61.9 | 87.3 | 83.7 | 77.1 | 71.2 | 67.3 | 3.87 | 61.9 | 87.3 | 83.7 | 77.1 | 71.2 | 67.3 |
| | GPTQ | $0.13_{0.0}$ | 15.3 | 26.5 | 35.6 | 53.0 | 49.3 | 24.3 | 28.2 | 4.72 | 52.4 | 69.3 | 79.1 | 74.4 | 58.4 | 44.8 |
| | GPTVQ | $0.13_{0.0}$ | 5.28 | 42.0 | 71.6 | 75.9 | 66.5 | 58.9 | 55.4 | 4.27 | 54.9 | 72.9 | 82.6 | 74.8 | 65.9 | 63.1 |
| | MOEQ | $0.0_0$ | 13.4 | 38.9 | 49.8 | 60.3 | 49.9 | 44.2 | 40.5 | 5.45 | 57.4 | 80.2 | 78.9 | 71.9 | 63.4 | 60.1 |
| | LOPRO | $0.21_{0.08}$ | 5.01 | 55.3 | 82.5 | 80.6 | 74.9 | 63.7 | 60.2 | 4.13 | 60.3 | 85.9 | 82.7 | 76.2 | 69.2 | 66.2 |
| | TILEQ$_s$ | **0.16**$_{0.03}$ | 4.98 | 55.5 | 82.8 | **80.9** | **75.1** | 63.8 | 60.3 | **4.10** | 60.4 | **86.1** | **82.9** | 76.4 | **69.4** | 66.4 |
| | TILEQ$_v$ | **0.16**$_{0.03}$ | **4.78** | **56.3** | **83.8** | 80.5 | 74.8 | **64.4** | **61.4** | 4.13 | **61.4** | **86.1** | 82.8 | **77.2** | 69.2 | **66.6** |
| QWEN1.5-MOE-A2.7B | FP16 | - | 6.79 | 41.6 | 72.9 | 79.6 | 69.1 | 61.2 | 59.3 | 6.79 | 41.6 | 72.9 | 79.6 | 69.1 | 61.2 | 59.3 |
| | GPTQ | $0.13_{0.0}$ | 12.5 | 29.4 | 42.7 | 50.2 | 53.2 | 25.2 | 30.1 | 7.98 | 33.4 | 64.3 | 77.1 | 64.5 | 51.5 | 52.2 |
| | GPTVQ | $0.13_{0.0}$ | 8.12 | 34.1 | 68.4 | 71.4 | 62.5 | 53.9 | 49.8 | 7.1 | 38.8 | 71.2 | 78.4 | 68.3 | 59.6 | 57.7 |
| | MOEQ | $0.0_0$ | 5e5 | 34.9 | 34.8 | 59.0 | 58.2 | 48.4 | 38.5 | 8.21 | 32.6 | 63.6 | 76.2 | 63.8 | 50.1 | 50.4 |
| | LOPRO | $0.43_{0.31}$ | 7.52 | 39.9 | 72.7 | 77.6 | 68.2 | 56.8 | 53.4 | 6.94 | 40.1 | 72.9 | 78.8 | **69.4** | 60.8 | 57.6 |
| | TILEQ$_s$ | **0.16**$_{0.04}$ | 7.56 | 39.6 | 72.5 | 77.8 | **68.9** | 57.5 | 54.1 | 6.94 | **40.3** | 72.9 | 79.1 | **69.4** | **61.0** | 58.0 |
| | TILEQ$_v$ | **0.16**$_{0.04}$ | **7.35** | **40.2** | **73.4** | **78.5** | 68.6 | **58.8** | **55.5** | **6.93** | 40.1 | **74.1** | **80.0** | 68.6 | 60.3 | **58.3** |
| QWEN3-30B-A3B | FP16 | - | 8.07 | 52.6 | 79.2 | 79.7 | 70.3 | 79.6 | 58.8 | 8.07 | 52.6 | 79.2 | 79.7 | 70.3 | 79.6 | 58.8 |
| | GPTQ | $0.13_{0.0}$ | 14.6 | 31.1 | 54.4 | 68.9 | 57.2 | 53.1 | 43.2 | 9.42 | 44.0 | 70.3 | 75.1 | 63.4 | 72.2 | 50.1 |
| | GPTVQ | $0.13_{0.0}$ | 11.8 | 34.1 | 58.5 | 70.7 | 61.2 | 61.5 | 47.9 | 8.7 | 47.9 | 73.1 | 76.5 | 66.2 | 76.1 | 54.1 |
| | MOEQ | $0.0_0$ | 3e4 | 26.6 | 29.8 | 50.5 | 49.8 | 41.4 | 28.5 | 28.1 | 37.8 | 30.6 | 59.4 | 55.3 | 60.1 | 43.2 |
| | LOPRO | $0.58_{0.46}$ | 11.1 | 34.4 | 58.3 | 71.5 | 62.9 | 62.9 | 48.0 | 8.57 | 50.2 | 75.4 | **77.4** | 68.7 | 77.7 | 56.1 |
| | TILEQ$_s$ | **0.16**$_{0.04}$ | 11.3 | 34.6 | 58.4 | 71.8 | 63.1 | 62.9 | 48.3 | 8.58 | 50.3 | 76.3 | 77.3 | **68.9** | **77.9** | **56.9** |
| | TILEQ$_v$ | **0.16**$_{0.04}$ | **10.1** | **42.2** | **70.4** | **74.1** | **65.7** | **71.3** | **50.8** | **8.64** | **50.8** | **76.6** | 77.0 | 68.2 | 77.3 | 56.8 |

reliance on complex codebooks at higher precision. Notably, both variants share the same low-rank method backbone—indicating that the core innovation lies not in the quantizer itself, but in the 2D-Tiling mechanism. This modular design allows users to flexibly choose: TILEQ$_s$ for **hardware-friendly, low-latency deployment**, and TILEQ$_v$ for ultra-low-bit applications demanding **peak accuracy**.

**Generalization and Scalability:** TILEQ excels across diverse MoE models with varying sparsity and scale. On the large-expert model Mixtral-8×7B, TILEQ$_v$ effectively reducing it to 4.78 and significantly boosting downstream task performance. On sparse models like Qwen3-MoE, TILEQ still works well, achieving 71.3 MMLU at 2-bit. Notably, its lightweight low-rank (only **0.04** bit) attains near-optimal accuracy, yielding an **8×** reduction in memory overhead compared to per-expert (**0.04 vs. 0.32**). In summary, re-

gardless of whether experts are "**big-and-dense**" or "**small-and-sparse**", TILEQ adaptively maintains high accuracy. This scalability positions TILEQ as a universal compression framework for next-generation sparse MoE models.

### 4.3. Inference latency

We evaluate inference latency in MoE block on 3 GPU architectures and A800 results are shown in Figure 3 and the proportion of each module in Qwen1.5-MoE is shown in Figure 4. Other results refer to Appendix D.1.

**Prefill.** During prefill, all tokens are processed simultaneously, leading to large, dense activation matrices that favor compute-bound, regular operations. TILEQ achieves minimal latency (**less than 5%**) by replacing irregular, per-expert sparse dispatch with a single structured GEMM. This trend holds consistently across model scales and batch sizes,

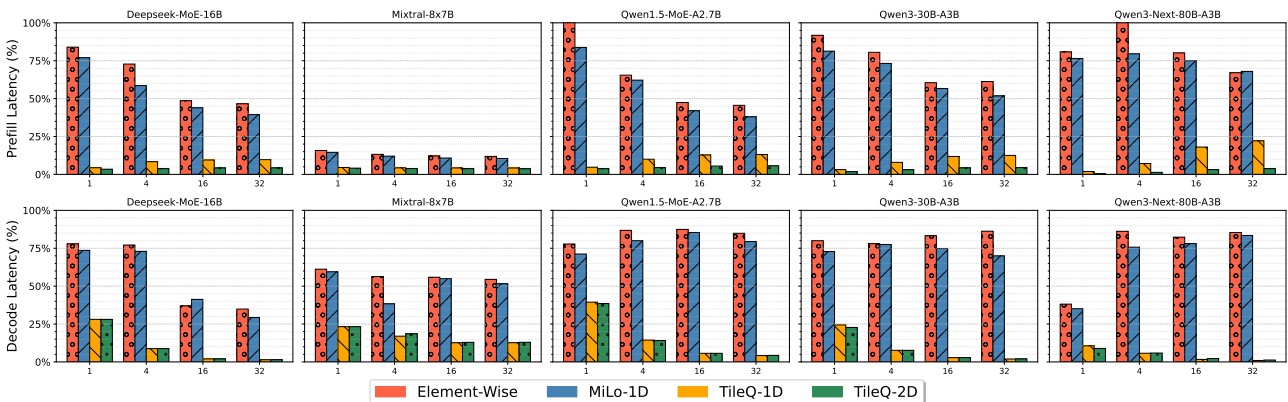

*Figure 3.* Inference latency in MoE MLP-block on A800. Here TILEQ-2D denotes our proposed 2D-tiling low-rank algorithm. The *Element-wise* baseline refers to the traditional per expert low-rank approach. The results of H800 and 5090 are given in Appendix D.1.

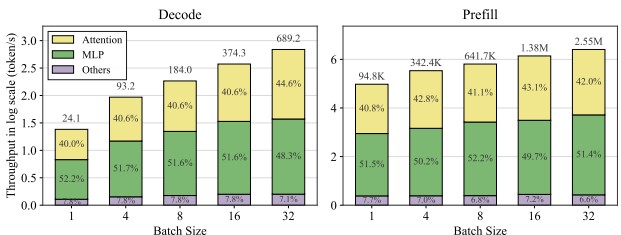

*Figure 4.* Inference throughput and the time proportion of each module in Qwen1.5-MoE-A2.7B with sequence length 4096.

confirming that TILEQ's 2D tiling maximizes hardware utilization during bulk processing.

**Decode.** Only few experts are active in this step, making memory bandwidth to be the primary bottleneck. Despite the reduced computational intensity, TILEQ remains a static, contiguous weight layout that enables efficient caching and vectorized access. Meanwhile, element-wise and Milo-1D are still slow (around 75% latency on Qwen3-MoE), underscoring its unsuitability for inference.

Thus, since the MoE MLP blocks account for about 50% of the total inference time in practice (Figure 4), the end-to-end latency of TILEQ is even lower in end-to-end deployment.

*Table 3.* Ablation studies on each component of TILEQ in QWEN1.5-MOE-A2.7B under 2-bit quantization.

| Ablation On | LORA. BITS | 2bit PPL↓ | 2bit ACC↑ | 3bit PPL↓ | 3bit ACC↑ |
|---|---|---|---|---|---|
| TILEQ$_v$ | 0.04 | **7.35** | **62.5** | **6.93** | **63.6** |
| ⌐× Vector $\mathcal{Q}$ | 0.04 | 7.56 | 61.6 | 6.94 | 63.5 |
| ⌐× Rotation | 0.04 | 7.6 | 61.8 | 6.97 | 63.5 |
| ⌐× 2D-Tiling | 0.31 | 7.49 | 61.6 | 6.94 | 63.5 |
| ⌐× LoRa | 0.00 | 12.5 | 38.5 | 7.98 | 57.2 |

### 4.4. Ablation Studies

To validate each component's contribution in TILEQ, we perform an ablation study, with results shown in Table 3.

TILEQ$_v$, incorporating 2D-Tiling, vector-wise quantization, and rotation, outperforms all variants at 2-bit, highlighting the synergy among these components. Replacing vector-wise with scalar quantization, or rotation, causes minor degradation (PPL to 7.56–7.60), suggesting limited benefits of higher-order optimizations at low bit-widths. However, at 3-bit, all configurations perform similarly, indicating higher bit-widths better suit advanced quantization strategies. Then, removing 2D-Tiling maintains accuracy but increases bit-width by **8×**. Conversely, eliminating the low-rank mechanism severely degrades performance—PPL rises to 12.5 and ACC drops to 38.5 at 2-bit. This confirms that leveraging expert similarity via 2D-Tiling with low-rank modeling is key for efficient compression, significantly enhancing accuracy without additional storage costs. Further ablation studies on rank and tile size are in Appendix D.1.

### 4.5. Quantization Time

As shown in Table 4 and the model names are in short of that in Table 1, the runtime of TILEQ is overwhelmingly dominated by the quantization phase, which accounts for more than 90% of the total time across all models. In contrast, the 2D-Tiling step is highly efficient by fast low-rank sketching, demonstrating its computational lightness and scalability. Notably, on sparse architectures such as Qwen3-30B-A3b, the quantization stage remains a bottleneck due to the inefficiency of GPTQ when applied to sparse experts and remains for future optimization.

*Table 4.* The time costs (hours) of each part in TILEQ

| Models | Total | Tiling | Quant |
|---|---|---|---|
| DS$_{16B}$ | 1.60 | 0.10 | 1.50 |
| M8$_{7B}$ | 2.22 | 0.22 | 2.00 |
| Q1.5$_{13B}$ | 1.52 | 0.12 | 1.40 |
| Q3$_{30B}$ | 5.70 | 0.20 | 5.50 |

# 5. Conclusion

In this work, we have presented TILEQ, an efficient low-rank PTQ method for MoE models. By introducing a novel 2D tiling strategy, TileQ significantly reduces the storage overhead of low-rank matrices while preserving model accuracy. Moreover, the structured computation pattern enabled by tiling facilitates highly parallel inference, leading to substantial reductions in latency compared to traditional MoE low-rank computation mechanisms. Overall, TILEQ serves as a **lightweight, general-purpose** extension to existing PTQ methods (e.g., GPTQ), achieving **SOTA performance** with **minimal memory overhead** and **negligible inference latency**, providing a **scalable** and **hardware-friendly** solution towards deploying MoE models under memory and computational constraints.

# Acknowledgements

This work is partially supported by the National Key R&D Program of China (E4YF620501) and Infinigence AI.

# Impact Statement

This paper presents work whose goal is to advance the field of machine learning. There are many potential societal consequences of our work, none of which we feel must be specifically highlighted here.

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

## Appendix Contents

**Organization:** In this appendix, we provide further details as follows:

*Table 5.* Symbols and Description in this paper.

| Symbols | Description |
|---|---|
| **Scalars** | |
| $K$ | Number of experts in each MoE module. |
| $i, o$ | The dimensions of input and output. |
| $r$ | Rank of the low-rank approximation. |
| $\mathcal{K}$ | Number of activated experts. |
| $r_0$ | Rank used in SVD of scaled weight (implied from $U_k, S_k, V_k$). |
| $M, N$ | Number of row/column tiles (implied from $U, V$ definitions). |
| $b, t$ | Token batch index and expert rank index (from $g_{b,t}$). |
| **Vectors** | |
| $u_k, v_k$ | Normalized vectorized $U_k$ and $V_k$ for clustering (Eq. 7). |
| $S \in \mathbb{R}^i$ | Gaussian sketch vector used in fast low-rank approximation. |
| **Matrices** | |
| $\mathcal{W} = \{W_k \in \mathbb{R}^{o \times i}\}_{k=1}^K$ | Set of expert weight matrices, each of size output $\times$ input. |
| $X \in \mathbb{R}^{B \times i}$ | Input batch tensor with $B$ tokens and $i$ input features. |
| $W_k$ | Weight matrix of the $k$-th expert, $W_k \in \mathbb{R}^{o \times i}$. |
| $U = [U_1, \ldots, U_M]^\top \in \mathbb{R}^{(Mi) \times r}$ | Stacked shared left low-rank factor across row tiles. |
| $V = [V_1, \ldots, V_N] \in \mathbb{R}^{r \times (No)}$ | Stacked shared right low-rank factor across column tiles. |
| $R$ | Residual matrix for expert $k$, i.e., $R = W_k - \tilde{W}_k$. |
| $\hat{W}_k$ | Quantized residual matrix for expert $k$, i.e., $\hat{W}_k = \mathrm{Quant}(R)$. |
| $s_k$ | Scaling diagonal matrix from calibration. |
| $\tilde{W}_k^{scale} = W_k \cdot s_k$ | Scaled weight matrix of expert $k$ (Eq. 5). |
| $U_k, S_k, V_k$ | Rank-$r_0$ SVD factors of $\tilde{W}_k$ (Eq. 6). |
| $\tilde{W}_k = U_{m_k} S V_{n_k}^\top s_k^{-1}$ | Low-rank matrix of $\tilde{W}_k^{scale}$ (Eq. 5). |
| $\mathcal{E}_{m,n}$ | Basis matrix for routing low-rank construction with 1 at $(m, n)$. |
| $W_{\mathrm{big}}$ | Large tiled matrix formed by placing $\tilde{W}_k$ into a 2D grid (Eq. 11). |
| $X_{\mathrm{proj}}$ | Projected input via shared left factor $US$. |
| $S \in \mathbb{R}^{b \times (Nr)}$ | Compact accumulator buffer avoiding 3D tensor (Eq. 16). |
| $Q$ | Intermediate matrix in sketch low-rank approximation (Eq. 13). |
| **Functions** | |
| $G(X)$ | Router function that outputs routing scores for experts. |
| $\mathrm{MoE}(X)$ | Output of the Mixture-of-Experts layer (Eq. 3). |
| $e_{k,X}$ | Set of top-$\mathcal{K}$ experts selected by the router for input $X$. |
| $\phi_k \mapsto \{p, q\}$ | Mapping from expert index $k$ to position $(p, q)$ in the 2D tile. |
| $\mathcal{Q}(\cdot)$ | Generic quantization function applied to the residual. |
| $\mathrm{QMoE}(X)$ | Standard quantized MoE forward pass using $\hat{W}_k$. |
| $\mathrm{LoTileMoE}(X)$ | Efficient low-rank inference path using shared $U, V$ and tiling (Eq. 15). |
| $\mathcal{G}_m^{\mathrm{row}}, \mathcal{G}_n^{\mathrm{col}}$ | Row and column expert clusters from biclustering (Eq. 8). |
| $\phi(k) = (m, n)$ | Final placement function assigning expert $k$ to a unique tile (Eq. 9). |
| $g_{b,t}$ | Routing weight for token $b$ and its $t$-th selected expert. |

# A. Detail of TILEQ

In this appendix, we provide a comprehensive theoretical and empirical analysis of TILEQ from three critical perspectives: compression ratio, quantization error, and inference latency. These analyses collectively justify the design choices in our method and explain its superiority over existing low-rank post-training quantization (PTQ) approaches for Mixture-of-Experts (MoE) models.

## A.1. Codes of TILEQ

**Quantization processes**    Algorithm 1 outlines process of MoE experts quantization in TILEQ.

---

**Algorithm 1** TileQ: 2D-Tiling Structured Low-Rank Quantization for MoE Models

---

**Require:** $\mathcal{W} = \{\boldsymbol{W}_k \in \mathbb{R}^{o \times i}\}_{k=1}^{K}$ (MoE expert weights), $Sample$ (calibration data), $d$ (quantization bit), $r$ (low-rank), $M, N$ (tiling grid size)

**Ensure:** $QModule$ (quantized MoE module), $\{\boldsymbol{U}_m\}_{m=0}^{M-1}, \{\boldsymbol{V}_n\}_{n=0}^{N-1}, \boldsymbol{\Sigma}$ (shared low-rank factors)

1: Initialize $QModule \leftarrow \emptyset$
2: **for** each MoE layer in Module **do**
3:     Collect input activations over calibration set: $\boldsymbol{X}_k \leftarrow$ forward($Sample$) for each expert $k$
       {Step 1: Scaling}
4:     **for** $k = 1$ **to** $K$ **do**
5:         Compute scaling vector: $s_k \leftarrow \overline{\boldsymbol{X}}_k^p / \sqrt{\max(\overline{\boldsymbol{X}}_k) \cdot \min(\overline{\boldsymbol{X}}_k)}$
6:         Scale weight: $\widetilde{\boldsymbol{W}}_k \leftarrow \boldsymbol{W}_k \cdot \mathrm{diag}(s_k)$
7:     **end for**
       {Step 2: Extracting low-rank features via sketching}
8:     Initialize $\mathcal{U} = [], \mathcal{V} = []$
9:     **for** $k = 1$ **to** $K$ **do**
10:        $\boldsymbol{U}_k, \boldsymbol{V}_k, \boldsymbol{\Sigma}_k \leftarrow$ SketchSVD($\widetilde{\boldsymbol{W}}_k^\top, r_0 = r/2$) {using Gaussian sketch, Eq. 13}
11:        Flatten and normalize: $u_k \leftarrow \mathrm{vec}(\boldsymbol{U}_k)/\|\mathrm{vec}(\boldsymbol{U}_k)\|, v_k \leftarrow \mathrm{vec}(\boldsymbol{V}_k)/\|\mathrm{vec}(\boldsymbol{V}_k)\|$
12:        Append to lists: $\mathcal{U}.append(u_k), \mathcal{V}.append(v_k)$
13:     **end for**
       {Step 3: Biclustering for 2D tiling assignment}
14:     Row clusters: $\{\mathcal{G}_m^{\mathrm{row}}\}_{m=0}^{M-1} \leftarrow$ KMeans($\mathcal{U}, n\_clusters = M$)
15:     Col clusters: $\{\mathcal{G}_n^{\mathrm{col}}\}_{n=0}^{N-1} \leftarrow$ KMeans($\mathcal{V}, n\_clusters = N$)
16:     Assign ideal tile: $(m_k^\star, n_k^\star) \leftarrow (\mathrm{cluster\_id}_U(k), \mathrm{cluster\_id}_V(k))$
       {Step 4: Greedy spatial placement to resolve collisions}
17:     $\phi \leftarrow$ GreedyPlacement($\{(m_k^\star, n_k^\star)\}_{k=1}^{K}, M, N$) {minimize $\ell_1$ distance, Eq. 9}
18:     Construct tiled matrix: $\boldsymbol{W}_{\mathrm{big}} \leftarrow \sum_{k=1}^{K} \mathcal{E}_{\phi(k)} \otimes \widetilde{\boldsymbol{W}}_k \in \mathbb{R}^{Mi \times No}$
       {Step 5: Global low-rank decomposition + residual quantization}
19:     $\boldsymbol{U}_{\mathrm{big}}, \boldsymbol{\Sigma}, \boldsymbol{V}_{\mathrm{big}} \leftarrow$ SketchSVD($\boldsymbol{W}_{\mathrm{big}}, r$)
20:     **for** $k = 1$ **to** $K$ **do**
21:        $(m, n) \leftarrow \phi(k)$
22:        Extract shared factors: $\boldsymbol{U}_k \leftarrow \boldsymbol{U}_{\mathrm{big}}[mi : (m+1)i, :], \boldsymbol{V}_k \leftarrow \boldsymbol{V}_{\mathrm{big}}[:, no : (n+1)o]$
23:        Reconstruct low-rank part: $\widetilde{\boldsymbol{W}}_k^{\mathrm{lr}} \leftarrow \boldsymbol{U}_k \boldsymbol{\Sigma} \boldsymbol{V}_k^\top \cdot \mathrm{diag}(s_k)^{-1}$
24:        Compute residual: $\boldsymbol{R}_k \leftarrow \boldsymbol{W}_k - \widetilde{\boldsymbol{W}}_k^{\mathrm{lr}}$
       {Quantize residual with Hessian-aware loss}
25:        $\boldsymbol{H}_k \leftarrow \mathbb{E}[\boldsymbol{X}_k \boldsymbol{X}_k^\top]$
26:        $\boldsymbol{W}_{k,q} \leftarrow \mathcal{Q}_{\mathrm{GPTQ/GPTVQ}}(\boldsymbol{R}_k; \boldsymbol{H}_k, d)$
27:        Store result: $QModule.\mathrm{expert}[k] \leftarrow (\boldsymbol{W}_{k,q}, \boldsymbol{U}_k, \boldsymbol{V}_k, \boldsymbol{\Sigma}, s_k, \phi(k))$
28:     **end for**
29: **end for**
30: **return** $QModule$

---

**Inference processes** Figure 1 shows the inference code of low-rank MoE layers in TILEQ.

```
1   # TileQ low-rank (MoE) layer.
2   class LoTileMoELinear(nn.Module):
3       def __init__(self, in_features, out_features, rank, row, col):
4           super().__init__()
5           self.in_features, self.out_features = in_features, out_features
6           self.rank, self.row, self.col = rank, row, col
7           self.U = nn.Parameter(torch.randn(in_features * row, rank))
8           self.V = nn.Parameter(torch.randn(rank, out_features * col))
9           self.S = nn.Parameter(torch.randn(rank))
10      def forward(self, x, routing_weights, selected_experts, expert_phi):
11          """
12          Args:
13              x , routing_weights, selected_experts: input, top-k weights, expert IDs.
14              expert_phi: 2D tile positions of all experts, shape (num_experts, 2).
15          Returns:
16              Output of shape (batch_seq, out_features).
17          """
18          batch_seq, in_features = x.shape
19          topk = selected_experts.size(1)
20          # Step 1: Compute U * S and project input x
21          us_weight = self.U.data * self.S.unsqueeze(0)
22          us_reshaped = us_weight.view(in_features, self.row * self.rank)
23          projected_x = torch.matmul(x, us_reshaped).view(bs, self.row, self.rank)
24
25          # Step 2: Gather projections using 2D expert positions
26          expert_coords = expert_phi[selected_experts]  # (batch_seq, topk, 2)
27          src_blocks = expert_coords[..., 0]  # (batch_seq, topk)
28          tgt_blocks = expert_coords[..., 1]  # (batch_seq, topk)
29          selected_projs = torch.gather(
30              projected_x, dim=1,
31              index=src_blocks.unsqueeze(-1).expand(-1, -1, self.rank)
32          )  # (bs, topk, rank)
33          weighted_projs = selected_projs * routing_weights.unsqueeze(-1) # (B, K, R)
34
35          # Step 3: Scatter into output accumulator by target block
36          expanded_tgt = tgt_blocks.unsqueeze(-1).expand(-1, -1, self.rank) # (B, K, R)
37          acc = torch.zeros(batch_seq, self.col, self.rank)
38          acc.scatter_add_(dim=1, index=expanded_tgt, src=weighted_projs)
39          flat_output = acc.view(batch_seq, self.col * self.rank)
40
41          # Step 4: Final projection | reshape V to (col * rank, out_features)
42          v_mat = ( self.V.data .reshape(self.rank, self.col, self.out_features)
43              .permute(1, 0, 2).reshape(self.col * self.rank, self.out_features) )
44          result = flat_output @ v_mat  # (batch_seq, out_features)
45          return result
```

*Figure 5.* Implementation of inference codes in TILEQ.

## A.2. Error bound

---

**Algorithm Review**

TileQ seeks a shared low-rank factorization that jointly approximates all experts in a 2D tiled layout. These scaled experts are then embedded into a large block matrix $\boldsymbol{W}_{\text{big}} \in \mathbb{R}^{Mi \times No}$ according to a placement mapping $\phi(k) = (p_k, q_k)$:

$$[\boldsymbol{W}_{\text{big}}]_{(p,q)} = \begin{cases} \tilde{\boldsymbol{W}}_k, & \text{if } \phi(k) = (p, q), \\ \mathbf{0}, & \text{otherwise,} \end{cases}$$

The overall optimization objective of TILEQ balances four key sources of approximation error:

1. **Per-module reconstruction error**: $\sum_{k=1}^{K} \|\boldsymbol{W}_k - \tilde{\boldsymbol{W}}_k^{\text{tile}}\|_F^2$ penalizes deviations between each original expert weight $\boldsymbol{W}_k$ and its reconstructed version from the tiled low-rank components. This term preserves individual expert functionality despite parameter sharing.

2. **Structural consistency**: $\phi$ enforces spatial coherence in the 2D tiling layout—e.g., by discouraging large displacements from ideal cluster-assigned positions $(m_k, n_k)$. This regularizer maintains the semantic alignment between expert similarity and tile placement.

3. **Global low-rank error**: $\|\boldsymbol{W}_{\text{big}} - \boldsymbol{U}\boldsymbol{\Sigma}\boldsymbol{V}^\top\|_F^2$ measures the fidelity of the shared 2D-tiling low-rank approximation across all experts. Minimizing this term ensures that the structured decomposition captures the dominant subspace of the aggregated expert weights.

4. **Quantization error**: $\sum_{k=1}^{K} \|\boldsymbol{R}_k - \mathcal{Q}(\boldsymbol{R}_k)\|_F^2$, $\boldsymbol{R}_k = \boldsymbol{W}_k - \tilde{\boldsymbol{W}}_k$ accounts for the distortion introduced when mapping full-precision weights to discrete quantized values.

---

From the review of TILEQ, the core optimization balances global compression, per-expert fidelity, structural coherence, and quantization compatibility. In the steps, errors do not accumulate progressively; instead, the error introduced at one stage is compensated for by the subsequent stage. Consequently, the final error of the algorithm is determined solely by the quantization error, which can be formalized as:

$$\mathcal{L}_{\text{TileQ}} = \underbrace{\sum_{k=1}^{K} \|\boldsymbol{R}_k - Q(\boldsymbol{R}_k)\|_F^2}_{\text{quantization error}} \leq \min_{\substack{\boldsymbol{W}_{\text{big}} = [\tilde{\boldsymbol{W}}_k]_{\phi(k)} \\ \boldsymbol{U}, \boldsymbol{\Sigma}, \boldsymbol{V}}} \underbrace{\|\boldsymbol{W}_{\text{big}} - \boldsymbol{U}\boldsymbol{\Sigma}\boldsymbol{V}^\top\|_F^2}_{\text{global low-rank error}}, \tag{19}$$

To capture the impact on model output, we adopt the activation-weighted norm induced by $\mathbf{A} = \mathbf{X}^\top \mathbf{X}$, which reflects how errors propagate to the actual predictions. Our goal is to bound the per-expert reconstruction error of this procedure.

**Error-Controlled Subspace Sharing via Activation-Aware Clustering** We now justify the local low-rank structure of experts in a manner that explicitly accounts for input activation statistics through $\mathbf{A} = \mathbf{X}^\top \mathbf{X}$. The error norm is $\mathcal{L}_{\text{TileQ}} = \text{tr}\left((\mathbf{W}_k - \hat{\mathbf{W}}_k)^\top \mathbf{A}(\mathbf{W}_k - \hat{\mathbf{W}}_k)\right)$.

For each expert $k$, compute the rank-$r$ approximation minimizing the $\mathbf{A}$-weighted error:

$$\tilde{\mathbf{W}}_k = \min_{\text{rank}(\mathbf{U}) \leq r} \|\mathbf{W}_k - \mathbf{U}\mathbf{V}^\top\|_\mathbf{A}^2. \tag{20}$$

This yields a generalized SVD under metric $\mathbf{A}$: $\tilde{\mathbf{W}}_k = \mathbf{U}_k \mathbf{S}_k \mathbf{V}_k^\top$, where columns of $\mathbf{U}_k \in \mathbb{R}^{m \times r}$ are $\mathbf{A}$-orthonormal: $\mathbf{U}_k^\top \mathbf{A} \mathbf{U}_k = \mathbf{I}_r$.

Define the *activation-aware left embedding* as the vectorized $\mathbf{A}$-normalized basis:

$$u_k = \frac{\text{vec}(\mathbf{U}_k)}{\|\text{vec}(\mathbf{U}_k)\|_{\mathbf{A} \otimes \mathbf{I}}} \in \mathbb{R}^{mr}, \tag{21}$$

where the norm is induced by the Kronecker product metric $\mathbf{A} \otimes \mathbf{I}_r$, i.e.,

$$\| \operatorname{vec}(\mathbf{U}) \|_{\mathbf{A} \otimes \mathbf{I}}^2 = \operatorname{tr}(\mathbf{U}^\top \mathbf{A} \mathbf{U}). \tag{22}$$

Similarly define $v_k$ from $\mathbf{V}_k$ (using output-side activations if available, or identity otherwise).

Apply KMeans to $\{u_k\}_{k=1}^K$ using Euclidean distance (which now implicitly respects $\mathbf{A}$ due to embedding construction). Let $c_m^U$ be the centroid of cluster $m$, and reshape it to matrix form $\bar{\mathbf{U}}_m$ such that $\operatorname{vec}(\bar{\mathbf{U}}_m) \propto c_m^U$ and $\bar{\mathbf{U}}_m^\top \mathbf{A} \bar{\mathbf{U}}_m = \mathbf{I}_r$ (via normalization).

By the $c$-approximation guarantee of KMeans (Kanungo et al., 2004),

$$\sum_{k=1}^K \| u_k - c_{m_k}^U \|_2^2 \le c \cdot \operatorname{OPT}_U, \tag{23}$$

where $\operatorname{OPT}_U$ is the optimal clustering cost.

Let $\mathcal{S}_k = \operatorname{col}(\mathbf{U}_k)$ and $\bar{\mathcal{S}}_m = \operatorname{col}(\bar{\mathbf{U}}_m)$ denote the column subspaces. The canonical angles between them under metric $\mathbf{A}$ satisfy the generalized Davis–Kahan bound (Yu et al., 2015):

$$\| \sin \Theta_{\mathbf{A}}(\mathcal{S}_k, \bar{\mathcal{S}}_{m_k}) \|_F \le \frac{\| \mathbf{U}_k - \bar{\mathbf{U}}_{m_k} \mathbf{Q} \|_{\mathbf{A}}}{\delta_k}, \tag{24}$$

for some orthogonal $\mathbf{Q} \in \mathbb{R}^{r \times r}$, where $\delta_k$ is the spectral gap between the $r$-th and $(r+1)$-th generalized singular values of $\mathbf{W}_k$ w.r.t. $\mathbf{A}$, assumed bounded below by $\delta > 0$. Crucially, both $\mathbf{U}_k$ and $\bar{\mathbf{U}}_{m_k}$ are $\mathbf{A}$-orthonormal, we have

$$\| \mathbf{U}_k - \bar{\mathbf{U}}_{m_k} \mathbf{Q} \|_{\mathbf{A}}^2 = 2r - 2 \operatorname{tr}\left( \mathbf{Q}^\top \bar{\mathbf{U}}_{m_k}^\top \mathbf{A} \mathbf{U}_k \right) \le C \cdot \| u_k - c_{m_k}^U \|_2^2, \tag{25}$$

for some constant $C$ depending on $r$ and norm alignment during centroid reshaping. Recall that in the Local Subspace Proximity model, we write

$$\tilde{\mathbf{W}}_k = \bar{\mathbf{U}}_{m_k} \mathbf{M}_k \bar{\mathbf{V}}_{n_k}^\top + \mathbf{E}_k, \tag{26}$$

and define $\epsilon_k = \| \mathbf{E}_k \|_{\mathbf{A}} = \sqrt{\operatorname{tr}(\mathbf{E}_k^\top \mathbf{A} \mathbf{E}_k)}$. Using the projection property of $\mathbf{A}$-orthonormal bases, the residual satisfies

$$\epsilon_k \le \| \tilde{\mathbf{W}}_k - \bar{\mathbf{U}}_{m_k}(\bar{\mathbf{U}}_{m_k}^\top \mathbf{A} \tilde{\mathbf{W}}_k) \|_{\mathbf{A}} \le \| \sin \Theta_{\mathbf{A}}(\mathcal{S}_k, \bar{\mathcal{S}}_{m_k}) \|_F \cdot \| \tilde{\mathbf{W}}_k \|_{\mathbf{A}}. \tag{27}$$

Combining with the Davis–Kahan bound and KMeans guarantee, we obtain

$$\epsilon_k \le \frac{\| \tilde{\mathbf{W}}_k \|_{\mathbf{A}}}{\delta} \cdot \sqrt{C} \cdot \| u_k - c_{m_k}^U \|_2. \tag{28}$$

Summing over all experts and applying the same argument to right subspaces, the total activation-weighted error is bounded by

$$\sum_{k=1}^K \epsilon_k^2 \le \frac{C'}{\delta^2} \left( \sum_{k=1}^K \| u_k - c_{m_k}^U \|_2^2 + \| v_k - c_{n_k}^V \|_2^2 \right) \le \frac{C'c}{\delta^2} \left( \operatorname{OPT}_U + \operatorname{OPT}_V \right). \tag{29}$$

Thus, the Local Subspace Proximity assumption is not heuristic: the per-expert error $\epsilon_k$ is provably controlled by the quality of activation-aware clustering and the stability of generalized singular subspaces under $\mathbf{A}$. When inputs induce strong similarity among expert subspaces (small $\operatorname{OPT}_U, \operatorname{OPT}_V$), the global quantization error $\| \mathbf{X}(\mathbf{W} - \hat{\mathbf{W}}) \|_F^2$ remains small.

**Reconstruction Error of TILEQ under Activation-Aware Subspace Sharing**   The analysis above establishes that when experts exhibit clustered activation-aware subspaces—i.e., small $\operatorname{OPT}_U$ and $\operatorname{OPT}_V$—their low-rank approximations can be well-approximated by shared bases within each cluster, with provably bounded residual error $\epsilon_k$. This insight directly motivates the design of TILEQ: by grouping experts into tiles based on their left/right singular vector embeddings (as constructed from $\mathbf{A} = \mathbf{X}^\top \mathbf{X}$), the method enforces exactly such local subspace sharing. Consequently, the tiling-induced approximation error $\epsilon_{\text{tiling}}^{(k)}$ inherits the guarantees derived above. We now formalize the total reconstruction error of TILEQ, combining this structured low-rank approximation with quantization.

**Theorem A.1** (Error of TILEQ). *Consider a Mixture-of-Experts (MoE) layer with* TILEQ *format. Then the error for expert $k$ satisfies*

$$\mathcal{E}^{(k)} = \|(\boldsymbol{W}_k - \tilde{\boldsymbol{W}}_k - \hat{\boldsymbol{W}}_k)\boldsymbol{X}\|_F \leq \mathcal{E}^{(k)}_{\mathrm{ind}} + \epsilon_{cluster} \tag{30}$$

*where $\mathcal{E}^{(k)}_{\mathrm{ind}}$ is the error of an independent per-expert low-rank + quantization baseline and $\epsilon_{cluster}$ is the error introduced by low-rank under 2D-Tiling.*

*Proof.* The total error is defined as

$$\mathcal{E}^{(k)} = \|(\boldsymbol{W}_k - \tilde{\boldsymbol{W}}_k - \hat{\boldsymbol{W}}_k)\boldsymbol{X}\|_F = \|(\boldsymbol{R}^{(k)} - \hat{\boldsymbol{W}}_k)\boldsymbol{X}\|_F. \tag{31}$$

First, consider the low-rank approximation step. The scaled weight $\tilde{\boldsymbol{W}}_k = \boldsymbol{W}_k \cdot \mathrm{diag}(s_k)$ admits a global SVD (or sketching) such that

$$\|\tilde{\boldsymbol{W}}_k - \boldsymbol{U}_k\boldsymbol{\Sigma}_k\boldsymbol{V}_k^\top\|_F \leq \epsilon_{\mathrm{svd}}. \tag{32}$$

Recovering in the unscaled domain gives

$$\|(\boldsymbol{W}_k - \boldsymbol{U}_k\boldsymbol{\Sigma}_k\boldsymbol{V}_k^\top \mathrm{diag}(s_k)^{-1})\boldsymbol{X}\|_F = \|\mathrm{diag}(s_k)^{-1}(\tilde{\boldsymbol{W}}_k - \boldsymbol{U}_k\boldsymbol{\Sigma}_k\boldsymbol{V}_k^\top)\boldsymbol{X}\|_F. \tag{33}$$

Let $s^{(k)}_{\min} = \min_i(s_k)_i > 0$. Then

$$\frac{1}{s^{(k)}_{\min}}\|(\tilde{\boldsymbol{W}}_k - \boldsymbol{U}_k\boldsymbol{\Sigma}_k\boldsymbol{V}_k^\top)\boldsymbol{X}\|_F \leq \frac{\epsilon_{\mathrm{svd}} \cdot \|\boldsymbol{X}\|_2}{s^{(k)}_{\min}}. \tag{34}$$

Thus, the low-rank error contribution is bounded by $\frac{\epsilon^{(k)}_{\mathrm{tiling}} \cdot \|\boldsymbol{X}\|_2}{s^{(k)}_{\min}}$, where $\epsilon^{(k)}_{\mathrm{tiling}} = \|\tilde{\boldsymbol{W}}_k - \boldsymbol{U}_k\boldsymbol{\Sigma}_k\boldsymbol{V}_k^\top\|_F$.

Second, for quantization, GPTQ guarantees

$$\|(\boldsymbol{R}^{(k)} - \hat{\boldsymbol{W}}_k)\boldsymbol{H}^{1/2}\|_F \leq \delta^{(k)}_{\mathrm{gptq}}. \tag{35}$$

Assuming i.i.d. token columns in $\boldsymbol{X}$, we have $\boldsymbol{H} = \mathbb{E}[\boldsymbol{X}\boldsymbol{X}^\top]$, and

$$\|(\boldsymbol{R}^{(k)} - \hat{\boldsymbol{W}}_k)\boldsymbol{X}\|_F^2 = \mathrm{Tr}\left((\boldsymbol{R}^{(k)} - \hat{\boldsymbol{W}}_k)\boldsymbol{H}(\boldsymbol{R}^{(k)} - \hat{\boldsymbol{W}}_k)^\top\right). \tag{36}$$

Since $\boldsymbol{H} \preceq \lambda_{\max}(\boldsymbol{H})\boldsymbol{I}$, it follows that

$$\|(\boldsymbol{R}^{(k)} - \hat{\boldsymbol{W}}_k)\boldsymbol{X}\|_F \leq \sqrt{\lambda_{\max}(\boldsymbol{H})} \cdot \|\boldsymbol{R}^{(k)} - \hat{\boldsymbol{W}}_k\|_F. \tag{37}$$

Moreover, the GPTQ guarantee implies

$$\|\boldsymbol{R}^{(k)} - \hat{\boldsymbol{W}}_k\|_F \lesssim \frac{\delta^{(k)}_{\mathrm{gptq}}}{\sqrt{\lambda_{\min}(\boldsymbol{H})}}, \tag{38}$$

hence

$$\|(\boldsymbol{R}^{(k)} - \hat{\boldsymbol{W}}_k)\boldsymbol{X}\|_F \lesssim \delta^{(k)}_{\mathrm{gptq}} \cdot \sqrt{\frac{\lambda_{\max}(\boldsymbol{H})}{\lambda_{\min}(\boldsymbol{H})}} = \delta^{(k)}_{\mathrm{gptq}} \cdot \kappa(\boldsymbol{H})^{1/2}. \tag{39}$$

Now compare with the independent baseline, where each expert uses its own SVD:

$$\tilde{\boldsymbol{W}}_k \approx \hat{\boldsymbol{U}}_k\hat{\boldsymbol{\Sigma}}_k\hat{\boldsymbol{V}}_k^\top, \quad \mathrm{rank}\ r_0, \tag{40}$$

with optimal error

$$\epsilon^{(k)}_{\mathrm{ind}} = \sum_{j=r_0+1}^{\min(o,i)} \sigma^{(k)}_j. \tag{41}$$

In contrast, the 2D-tiling method applies a global SVD on $\boldsymbol{W}_{\text{big}}$, yielding for any sub-block:

$$\|\tilde{\boldsymbol{W}}_k - \boldsymbol{U}_k \boldsymbol{\Sigma}_k \boldsymbol{V}_k^\top\|_F \leq \sum_{j=r+1}^{\min(Ro,Ci)} \sigma_j(\boldsymbol{W}_{\text{big}}). \tag{42}$$

Define the excess tiling error

$$\eta^{(k)} := \epsilon_{\text{tiling}}^{(k)} - \epsilon_{\text{ind}}^{(k)} \geq 0. \tag{43}$$

When experts exhibit clustered activation-aware singular vectors—i.e., when $\text{OPT}_U$ and $\text{OPT}_V$ are small—biclustering produces row/column clusters $\mathcal{G}_m^{\text{row}}, \mathcal{G}_n^{\text{col}}$, and the extra error from shared bases satisfies

$$\eta^{(k)} \leq O\left(\frac{1}{\sqrt{|\mathcal{G}_m^{\text{row}}|}} + \frac{1}{\sqrt{|\mathcal{G}_n^{\text{col}}|}}\right) \cdot \text{diam(cluster)}, \tag{44}$$

Combining both error sources, we obtain

$$\mathcal{E}^{(k)} \leq \underbrace{\frac{\epsilon_{\text{tiling}}^{(k)} \cdot \|\boldsymbol{X}\|_2}{s_{\min}^{(k)}}}_{\text{low-rank error}} + \underbrace{\delta_{\text{gptq}}^{(k)} \cdot \kappa(\boldsymbol{H})^{1/2}}_{\text{quantization error}}, \tag{45}$$

Recall from Equations (45) that both methods share the same quantization error term. Therefore,

$$\begin{aligned} \mathcal{E}^{(k)} &\leq \frac{\epsilon_{\text{tiling}}^{(k)} \cdot \|\boldsymbol{X}\|_2}{s_{\min}^{(k)}} + \delta_{\text{gptq}}^{(k)} \cdot \kappa(\boldsymbol{H})^{1/2} \\ &= \frac{(\epsilon_{\text{ind}}^{(k)} + \eta^{(k)}) \cdot \|\boldsymbol{X}\|_2}{s_{\min}^{(k)}} + \delta_{\text{gptq}}^{(k)} \cdot \kappa(\boldsymbol{H})^{1/2} \\ &= \mathcal{E}_{\text{ind}}^{(k)} + \frac{\eta^{(k)} \cdot \|\boldsymbol{X}\|_2}{s_{\min}^{(k)}} \end{aligned} \tag{46}$$

Let $\epsilon_{cluster} = \frac{\eta^{(k)} \cdot \|\boldsymbol{X}\|_2}{s_{\min}^{(k)}}$ as the error introduced by low-rank under 2D cluster tiling. we have

$$\mathcal{E}^{(k)} \leq \mathcal{E}_{\text{ind}}^{(k)} + \epsilon_{cluster} \tag{47}$$

$\square$

**Discussion**  The theoretical analysis establishes that the reconstruction fidelity of TILEQ hinges on two interrelated conditions:

1. **Global low-rank structure**: The block matrix $\boldsymbol{W}_{\text{big}}$ admits an accurate low-rank approximation when its singular values decay rapidly. This occurs precisely when experts exhibit strong similarity in their activation-aware subspaces—i.e., when the optimal clustering costs $\text{OPT}_U$ and $\text{OPT}_V$ are small—as formalized in ¶ A.2.

2. **Local subspace alignment**: The residual error $\epsilon_k$ quantifies the deviation of expert $k$ from the shared subspace assigned to its tile. By clustering experts based on their activation-aware left and right singular vectors ($u_k$, $v_k$), the biclustering step promotes small $\epsilon_k$, provided the underlying subspaces are well-separated.

Critically, TILEQ avoids the over-constrained sharing of prior approaches that enforce a single global $\boldsymbol{U}$ or $\boldsymbol{V}$. Instead, its 2D tiling enables *structured parameter sharing*: experts assigned to the same *row* of the tiling share a common left basis $\bar{\boldsymbol{U}}_m$ (capturing output-space structure), while those in the same *column* share a right basis $\bar{\boldsymbol{V}}_n$ (modeling input-space structure). This design preserves model expressivity while drastically reducing redundancy.

When the activation-aware singular subspaces of experts are well-clustered—a condition reflected by small $\text{OPT}_U$ and $\text{OPT}_V$, and commonly observed in practice due to routing-induced functional overlap—the per-expert reconstruction error of TILEQ nearly matches that of an independent low-rank baseline. Yet, it achieves this with only $O((M + N)r)$ shared basis matrices, compared to $O(Kr)$ for independent decompositions. This confirms TILEQ's suitability for scalable, high-fidelity quantization of large MoE models.

### A.3. Compression Ratio Analysis

In this part, we provide a analysis of the compression ratio on **(i)** element-wise quantization (Zhang et al., 2024a; Li et al., 2025b); **(ii)** 1D shared low-rank quantization (Huang et al., 2025; Li et al., 2025c) **(iii)** 2D-Tiling low-rank quantization (TILEQ).

Most definitions follow table 5. We denote the following parameters to derive the method compression efficiency:

- $d$: target quantization bit-width for quantized components (e.g., 2 or 3 bits),

- $d_{\text{fp}}$: bit-width of the original floating-point format (e.g., 16 for `float16`),

- $d_{\text{low}}$: bit-width used to store the shared low-rank factors in TILEQ $U, V$ (typically `fp8`, i.e., 8 bits),

- $g$: group size used in grouped quantization (for scaling factors).

**Basic Quantization.** This baseline applies standard post-training quantization (PTQ) directly to the full expert weights without any low-rank decomposition. Each weight matrix $W_k$ is quantized independently using either scalar or vector (group-wise) quantization. The storage cost consists of:

- $K \cdot oi \cdot d$ bits for quantized weights,

- $K \cdot \lceil \frac{oi}{g} \rceil \cdot d_{\text{fp}}$ bits for scaling factors (with $g = 1$ for scalar quantization and $g > 1$ for grouped/vector quantization).

Thus, the average bit-width per weight element is simply as $d_{\text{basic}} = d + \frac{d_{\text{fp}}}{g}$:

**Element-wise Low-Rank Quantization.** In this variant, each expert weight matrix $W_k$ is independently decomposed into a low-rank approximation without any parameter sharing across experts. The extra storage cost includes:

- $K \cdot (or + ir) \cdot d_{\text{low}}$ bits for all $U_k$ and $V_k$ factors,

- $K \cdot r^2 \cdot d_{\text{fp}}$ bits for the singular value matrices $\Sigma_k$.

Dividing by the total number of weight elements $Koi$, the average bit-width per element is:

$$d_{\text{per-expert}} = d_{\text{basic}} + \frac{(or + ir) \cdot d_{\text{fp}}}{oi} + \frac{r^2 \cdot d_{\text{fp}}}{oi} = d_{\text{basic}} + r d_{\text{fp}} \left( \frac{1}{i} + \frac{1}{o} \right) + \frac{r^2 d_{\text{fp}}}{oi}. \tag{48}$$

This approach incurs significantly higher low-rank storage overhead compared to both 1D and TILEQ due to the absence of any cross-expert sharing, especially when $K$ is large.

**1D Shared Low-Rank Quantization.** This baseline concatenates all expert weights along one dimension (e.g., $W_{\text{cat}} \in \mathbb{R}^{o \times (Ki)}$) and applies a shared left factor $U \in \mathbb{R}^{o \times r}$, yielding $W_{\text{cat}} \approx UV$. Each expert's low-rank component is reconstructed from slices of $V$. Low-rank storage includes:

- $or \cdot d_{\text{fp}}$ bits for the shared factor $U \in \mathbb{R}^{o \times r}$, with a total size of $or$,

- $Kir \cdot d_{\text{fp}}$ bits for $V \in \mathbb{R}^{r \times Ki}$, with no sharing across experts and a total size of $Kir$.

The total number of weight elements is $Koi$, so the average bit-width is:

$$d_{\text{1D}} = d_{\text{basic}} + (\frac{r}{Ki} + \frac{r}{o}) d_{\text{fp}}. \tag{49}$$

Note that the term $\frac{r}{o} d_{\text{fp}}$ arises from the shared $U$ amortized over all experts.

**TILEQ: 2D-Tiling Structured Low-Rank Quantization.** In TILEQ, experts are arranged into an $M \times N$ grid. The shared factors are:

- $\boldsymbol{U} = [\boldsymbol{U}_1^\top, \dots, \boldsymbol{U}_M^\top]^\top \in \mathbb{R}^{Mi \times r}$, composed of $M$ row blocks with a total size of $Mir$,

- $\boldsymbol{V} = [\boldsymbol{V}_1, \dots, \boldsymbol{V}_N] \in \mathbb{R}^{r \times No}$, composed of $N$ column blocks with a total size of $Nor$,

- Singular values $\boldsymbol{\Sigma} \in \mathbb{R}^{r \times r}$ (shared globally or per tile; we assume global for minimal overhead), with a total size of $r^2$ and stored in $r^2 \cdot d_{\text{fp}}$ bits.

Since $K \leq MN$, the average bit-width per weight element becomes:

$$d_{\text{TileQ}} = d_{\text{basic}} + \underbrace{\frac{Mir \cdot d_{\text{low}} + Nor \cdot d_{\text{low}}}{Koi}}_{\text{shared low-rank overhead}} + \underbrace{\frac{r^2 \cdot d_{\text{fp}}}{Koi}}_{\text{singular vector}} \approx d_{\text{basic}} + rd_{\text{low}}\left(\frac{M}{Ko} + \frac{N}{Ki}\right). \tag{50}$$

Under the conditions $M = N$, $i = o$, the extra bit-widths $d'$ of three quantization methods are:

$$d'_{\text{per-expert}} \approx \frac{2rd_{\text{fp}}}{i}, \quad d'_{\text{1D}} \approx \left(\frac{r}{i}\right)d_{\text{fp}}, \quad d'_{\text{TileQ}} \approx \frac{2rd_{\text{low}}}{i\sqrt{K}}, \tag{51}$$

where the singular value overhead in $d_{\text{TileQ}}$ is omitted as it is negligible for large $K$. From the results, TILEQ reduces the additional memory by $2\sqrt{K}$ compared with element-wise methods, and by $\sqrt{K}$ times compared with 1D-sharing mathods.

In summary, TILEQ achieves the lowest average bit-width among the three methods due to its two-dimensional parameter sharing, which exploits expert similarity more effectively and reduces redundancy in both input and output feature spaces simultaneously. This theoretical advantage directly translates into higher compression ratios and lower memory footprint at inference time.

---

**Example.** Numerical analyze on Qwen3-30B-A3B

We instantiate our analysis using a representative MoE layer from `Qwen3-30B-A3B`, with the following configuration:

- Expert weight: $\boldsymbol{W}_k \in \mathbb{R}^{o \times i} = \mathbb{R}^{2048 \times 768}$ (i.e., $i = 768$, $o = 2048$) with experts $K = 128$.

- Let quantization bit $d = 2$; groupsize $g = 128$ and approximation rank $r = 32$,

The base quantization cost is $d_{\text{basic}} = d + d_{\text{fp}}/g = 3 + 16/128 = 2.125$ bits per weight. Below, we report the *additional* average bit-width for each method.

- **Per-Expert Low-Rank:**

$$\Delta b_{\text{per-expert}} = rd_{\text{fp}}\left(\frac{1}{i} + \frac{1}{o}\right) + \frac{r^2 d_{\text{fp}}}{io} = 32 \cdot 8\left(\frac{1}{768} + \frac{1}{2048}\right) + \frac{32^2 \cdot 16}{768 \cdot 2048} \approx 0.417 + 0.010 = \mathbf{0.427} \text{ bits.}$$

- **1D Shared Low-Rank:**

$$\Delta b_{\text{1D}} = \frac{rd_{\text{fp}}}{Ki} + \frac{rd_{\text{low}}}{o} = \frac{32 \cdot 16}{128 \cdot 768} + \frac{32 \cdot 16}{2048} \approx 0.0052 + 0.256 = \mathbf{0.257} \text{ bits.}$$

- **TILEQ (with tiling $M = 16$ and $N = 8$):**

$$\Delta b_{\text{TileQ}} \approx rd_{\text{low}}\left(\frac{M}{Ko} + \frac{N}{Ki}\right) = 32 \cdot 8\left(\frac{8}{128 \cdot 2048} + \frac{8}{128 \cdot 768}\right) \approx 0.0117 + 0.0313 = \mathbf{0.041} \text{ bits.}$$

This example demonstrates that TILEQ reduces the low-rank storage overhead by **6×** compared to 1D sharing and by **10×** compared to per-expert decomposition and when the model becomes sparser (i.e., with a larger $K$), the advantage of TILEQ becomes more pronounced.

## A.4. Time Complexity

In TILEQ, the forward pass consists of the following key steps:

**1. Global Input Projection.** The input tensor $X \in \mathbb{R}^{B \times i}$ is multiplied with the reshaped shared factor $(U\Sigma)_{\text{reshape}} \in \mathbb{R}^{i \times (Mr)}$:

$$X_{\text{proj}} = X \cdot (U\Sigma)_{\text{reshape}} \in \mathbb{R}^{B \times (Mr)}. \tag{52}$$

This is a single dense GEMM with time complexity $\mathcal{O}(BiMr)$.

**2. Routing-Weighted Selection and Accumulation.** For each token–expert pair $(b, t)$, the algorithm:

- Extracts a rank-$r$ slice from $X_{\text{proj}}$ using precomputed tile coordinates $(m_{b,t}, n_{b,t})$,
- Scales it by the routing weight $g_{b,t}$,
- Accumulates the result into a buffer indexed by column tile $n_{b,t}$ via `scatter_add`.

The selection involves indexing into a $(B, Mr)$ tensor using $(B, \mathcal{K}, r)$ indices—costing $\mathcal{O}(B\mathcal{K}r)$ memory operations. The accumulation uses a vectorized `scatter_add` over $B\mathcal{K}r$ elements into a $(B, N, r)$ buffer, also $\mathcal{O}(B\mathcal{K}r)$.

**3. Output Reconstruction.** The accumulated buffer $X_{\text{sum}} \in \mathbb{R}^{B \times (Nr)}$ is multiplied with the reshaped shared right factor $V_{\text{flat}} \in \mathbb{R}^{(Nr) \times o}$:

$$Y = X_{\text{sum}} \cdot V_{\text{flat}} \in \mathbb{R}^{B \times o}, \tag{53}$$

which is another dense GEMM with complexity $\mathcal{O}(BNro)$.

**Total Time Complexity.** Summing the dominant terms, the overall time complexity of TILEQ inference is:

$$T_{\text{TileQ}} = \mathcal{O}\big(BiMr + BNro + B\mathcal{K}r\big). \tag{54}$$

---

> **Insights** ♀ Why is LOTILEMOE Fast?
>
> LOTILEMOE achieves low-latency inference by replacing irregular, sparse expert computation with structured, batched dense operations. We analyzed its efficiency in prefill–decode (P–D) separation serving through:
>
> - **vs. Per-Token Expert Dispatch.** In a MoE implementation with low-rank decomposition, each token–expert pair $(b, t)$ executes an independent small GEMM, which leads to $B\mathcal{K}$ separate kernel launches. During **prefill** (large $B$, compute-bound), this results in poor arithmetic intensity and wasted compute capacity. During **decode** (small $B = 1$, memory-bound), the overhead is even worse: frequent kernel launches, non-coalesced memory accesses, and scattered weight reads dominate latency. The total cost is:
>
> $$T_{\text{per-token}} = \mathcal{O}\big(B\mathcal{K}(ir + ro)\big) + \underbrace{\Omega(B\mathcal{K})}_{\text{kernel launch \& memory scatter}},$$
>
> where the second term reflects unavoidable software and hardware inefficiencies. Moreover, these GEMMs have highly unfavorable shapes—typically "tall-and-skinny" that cannot fully utilize modern accelerators (e.g. Tensor core, NPU). As a result, despite modest theoretical FLOP counts, per-token dispatch exhibits poor hardware efficiency and high latency.
>
> In contrast, LOTILEMOE performs only two large, well-shaped GEMMs for the entire batch—$\mathcal{O}(BiMr)$ and $\mathcal{O}(BNro)$—with inner dimensions $(Mr, Nr)$ large enough to activate tensor-core pipelines. Combined with vectorized indexing and scatter-add, this eliminates per-token overhead and fully exploits hardware parallelism.
>
> - **vs. 1D Shared Low-Rank.** Methods like MILO share a matrix across all experts but keep the other factor expert-specific. This corresponds to a degenerate 2D tiling configuration with $M = 1$ and $N = K$, i.e., no sharing along the output dimension. In low-rank computation, the bottleneck lies in loading and storing the weights in both prefill and decode stages, whereas LOTILEMOE reduces the costs significantly (e.g., **6× less in qwen3**) by balanced 2D tiling. This directly lead to higher effective bandwidth.

> Thus, LOTILEMOE is fast not because it reduces asymptotic FLOPs dramatically, but because it restructures computation to align with the strengths of modern accelerators—**dense linear algebra**, **minimal memory access**, and **efficient control flow**—while preserving cross-expert parameter sharing through 2D tiling.

### A.5. Memory Costs

We further analyze the inference-time memory cost of TileQ under different rank settings. As illustrated in Figure 2, TileQ does not materialize large intermediate matrices such as $G \times X_{tile}$ during inference after optimization. Instead, it performs efficient Scatter/Gather operations on smaller intermediate tensors, as shown in Figure 2c. Ignoring the index tensors and small low-rank matrices such as $U$ and $V$, the main additional intermediate tensors correspond to `selected_projs`, `weighted_projs`, and `expanded_tgt` in the implementation shown in Figure 5, whose dimensions are

$$batchsize \times seqlen \times topk \times rank.$$

By comparison, the activation tensor $X$ has dimensions

$$batchsize \times seqlen \times infeatures.$$

Since the first two dimensions are shared, the additional memory introduced by TileQ is mainly determined by $topk \times rank$, while the activation memory is determined by $infeatures$. In our experiments, $topk$ is typically smaller than 10, and a rank of 32 is sufficient to maintain model accuracy. Meanwhile, the minimum value of $infeatures$ among our evaluated models is 512. Therefore, the relative additional memory cost can be estimated as

$$\frac{topk \times rank}{infeatures} \leq \frac{10 \times 32}{512}.$$

Moreover, these intermediate tensors are layer-local and do not accumulate across layers during inference. Thus, the additional inference-time memory footprint introduced by TileQ remains limited.

## B. Evaluation Details

**Hardware Configuration.** All quantization and accuracy evaluations in our experiments were conducted on a single NVIDIA A800-80GB GPU, with the exception of downstream evaluations for the Mixtral-8x7B model, which required two A800 GPUs due to out-of-memory (OOM) errors on a single device. For inference latency measurements, we performed MLP block latency on 3 GPU platforms: A800, H800, and RTX 5090.

**Calibration Strategy.** Our calibration process utilizes a dataset comprising 128 randomly sampled sequences from the C4 corpus (Raffel et al., 2020), each containing 2048 tokens. This approach has been demonstrated to be effective in prior works such as OmniQuant (Shao et al., 2023) and AffineQuant (Ma et al., 2024).

**Experimental Settings.** In both perplexity and inference efficiency experiments, we uniformly set the sequence length to 4096 tokens across all models. For downstream task evaluation, we employed the `lm-eval` (Gao et al., 2024) framework consistently and reported raw accuracy (`acc`)—not normalized accuracy (`acc_norm`)—as the final metric and we adopt the $n-$shot ($ns$) param to each tasks from OpenLLM-V1 benchmark. Details are as follows:

- **ARC-Challenge (AC)** and **ARC-Easy (AE),** $ns = 0$ (Boratko et al., 2018): The AI2 Reasoning Challenge (ARC) consists of grade-school-level multiple-choice science questions. ARC-Challenge focuses on questions that are resistant to simple retrieval or co-occurrence heuristics and demand genuine reasoning, whereas ARC-Easy contains items more solvable by basic methods.

- **PIQA (QA),** $ns = 0$ (Bisk et al., 2020): The Physical Interaction Question Answering dataset assesses a model's grasp of physical commonsense by asking it to select the more plausible outcome between two scenarios involving everyday object interactions.

- **Winogrande (WI),** $ns = 0$ (Sakaguchi et al., 2021): Designed to scale up the Winograd Schema Challenge, Winogrande uses adversarial filtering to construct difficult pronoun resolution problems that require deep contextual understanding.

- **Hellaswag (HS),** $ns = 10$ (Zellers et al., 2019): This tests commonsense inference through sentence completion, where models choose the most likely continuation of a scenario from multiple options, including adversarially crafted distractors.

- **Massive Multitask Language Understanding (MU),** $ns = 5$ (Hendrycks et al., 2020): The task spans 57 subjects across STEM, humanities, and social sciences, measuring both factual knowledge and the capacity for reasoning in specialized domains beyond superficial pattern matching.

**Hyperparameter selection.** The hyperparameters in TileQ are specified as follows:

- **Tile-Rank**: the rank of the low-rank approximation in the 2D-tiling matrix (see Figure 1), setting to **32**.

- **Tiling-size**: the dimensions $(M, N)$ of the tiling grid, which determines how the $K$ experts are arranged into an $m \times n$ block matrix. We choose $m = \lfloor \sqrt{K} \rfloor$ (the integer closest to $\sqrt{K}$) and set $n = \lceil K/m \rceil$ and ensure $M * N \geq K$, this ensuring a near-square layout that balances row and column coherence while accommodating all experts.

**Quantization Implementation.** In the quantization phase of TileQ, we adopt a multi-level quantization strategy. Specifically, for scalar and vector quantization, we employ the Hessian-based proxy quantization scheme from GPTQ (Frantar et al., 2023) and GPTVQ (Van Baalen et al., 2024), without incorporating advanced optimizations such as weight clipping or learnable codebooks. For the rotation component, our approach aligns with the rotation technique used in LoPRo (Gu et al., 2026).

**Baseline Methods.** We primarily compare against three established baselines: GPTQ (Frantar et al., 2023), GPTVQ (Van Baalen et al., 2024), and LoPRo (Gu et al., 2026) where LoPRo represents the most recent state-of-the-art method for ultra-low-bit (e.g., 2–3 bit) weight-only PTQ that we are aware of, consistently outperforming earlier approaches such as QuIP# (Tseng et al., 2024a), OmniQuant (Shao et al., 2023), and others in terms of accuracy.

Other low-rank PTQs such as SVDQuant (Li et al., 2025b) and LQER (Zhang et al., 2024a)—have been proposed, and several works target MoE models specifically (e.g., MoEQuant (Chen et al., 2025c), MiLo (Huang et al., 2025)), we found it impractical to include them in a fully fair and comprehensive comparison across all 4 evaluation models. The main obstacles include unavailability of open-source code, lack of support for newer MoE architectures, absence of 2–3 bit PTQ configurations, or insufficient experimental details necessary for faithful reproduction. Consequently, we restrict our main accuracy comparison to the three aforementioned methods listed in Table 1. In inference efficiency evaluations, only MiLo and SVDQuant focused on efficient low-rank inference: MiLo aims at MoE models, while SVDQuant targets diffusion models. So we adopt the low-rank MoE design from MiLo for latency evaluations.

# C. More Ablation Studies

In this section, we validate the impact and justify the final selection of two key hyperparameters in TILEQ by scalar quantizing the Qwen1.5-MoE-A2.7B model to 2-bit.

## C.1. Ablation on Rank

*Table 6.* Ablation on rank of Qwen1.5-MoE-A2.7B in 2bit TILEQ$_S$, where Avg.Rank stands for the rank evenly distributed to each expert. Latency was tested in batch size 16. We set the rank 32 as the final setting in main evaluations.

| Tile-Rank | Latency | | Memory | | Performance | | | | | | |
|---|---|---|---|---|---|---|---|---|---|---|---|
| | Decode | Prefill | Avg.Rank | Ex.Bit | Wiki | AC | AE | PQ | WI | MU | HS |
| 8 | 3.6% | 2.3% | 0.08 | 0.01 | 7.62 | 36.6 | 69.2 | 77.1 | 68.9 | 56.6 | 53.7 |
| 16 | 4.2% | 3.4% | 1.63 | 0.03 | 7.56 | 37.8 | 69.7 | 77.8 | 69.2 | 57.3 | 53.9 |
| 32 | 5.0% | 4.2% | 3.26 | 0.06 | 7.54 | 37.5 | 70.8 | 77.6 | 68.9 | 57.5 | 54.1 |
| 64 | 7.1% | 7.5% | 6.5 | 0.11 | 7.53 | 37.8 | 70.6 | 78.1 | 68.9 | 57.7 | 54.4 |
| 128 | 12.1% | 10.3% | 12.7 | 0.22 | 7.52 | 38.6 | 71.5 | 78.3 | 69 | 67.8 | 53.8 |

As shown in Table 6, we conduct an ablation study on the TileQ rank (denoted as *Tile-Rank*) to investigate its impact on model performance, memory overhead, and inference latency in Qwen1.5-MoE-A2.7B. The results reveal several key observations.

Firstly, increasing the Tile-Rank consistently improves model accuracy across most benchmarks. This suggests that higher-rank approximations better preserve the expressive capacity of the original experts. However, despite the improvement in accuracy, the marginal gains diminish beyond rank 32 on many metrics, indicating a saturation point where additional rank yields diminishing returns. Secondly, both memory overhead (measured in average rank per expert and extra bits for factor storage) and decoding latency scale sublinearly with Tile-Rank, which bring a huge burden in memory cost and latency at a big rank (64 or more).

In summary, we select **Tile-Rank = 32** as our default configuration, which strikes a favorable trade-off among accuracy, memory costs, and inference speed. At this setting, TileQ achieves near-peak performance on most general reasoning and commonsense tasks while maintaining low overhead—validating that structured low-rank sharing can effectively compress MoE models without sacrificing practical utility

### C.2. Ablation on Tile

Table 7 presents an ablation study on the 2D tiling configuration $(M, N)$, which governs how the 60 experts are arranged into a block matrix for shared low-rank approximation. We observe a clear U-shaped trade-off between inference latency and model performance with respect to layout aspect ratio. Extremely skewed layouts (e.g., $1 \times 60$ or $60 \times 1$) suffer from high decoding latency ($>13\%$) due to poor memory access patterns and limited cross-expert coherence in one dimension. Moreover, when the tiling dimension $M$ are large, the decoding latency increases significantly. This is because a larger $M$ leads to a taller shared left factor matrix $\mathbf{U}$, which in turn requires more extensive *gather-scatter* operations (Figure 5: line 29,38) during expert reconstruction, which is memory-bound and suffers from poor cache locality and limited vectorization efficiency. Consequently, as $M$ grows, the proportion of time spent on these unstructured memory operations rises, degrading overall inference throughput—even though the total number of parameters and arithmetic operations remains unchanged. This explains why layouts with large $M$ (e.g., $32 \times 2$, $60 \times 1$) exhibit higher latency despite using the same Tile-Rank.

In contrast, near-square tilings—particularly $8 \times 8$ and $16 \times 4$—achieve the best balance: they minimize decode latency (as low as 5.1%) while delivering peak or near-peak accuracy across most benchmarks (e.g., ARC Challenge reaches 39.6 at $8 \times 8$). This confirms that balanced 2D clustering enhances both hardware efficiency and representation sharing. Consequently, we adopt the $8 \times 8$ layout as our default, which provides the lowest latency without sacrificing task performance.

*Table 7.* Ablation study on tiling dimensions $(M, N)$ in TileQ for Qwen1.5-MoE-A2.7B (2-bit). $M$ and $N$ denote the number of rows and columns in the 2D expert layout ($M \times N \geq K = 60$). Latency is reported with batch size 16.

| Tile-Dimention | | Latency | | Performance | | | | | | |
|---|---|---|---|---|---|---|---|---|---|---|
| M | N | Decode | Prefill | Wiki | AC | AE | PQ | WI | MU | HS |
| 1 | 60 | 13.2% | 5.4% | 7.62 | 38.1 | 70.2 | 76.8 | 70.0 | 58.4 | 53.9 |
| 2 | 30 | 10.1% | 5.3% | 7.65 | 37.8 | 71.2 | 77.5 | 68.7 | 57.2 | 53.9 |
| 4 | 15 | 7.8% | 5.2% | 7.56 | 37.2 | 71.2 | 77.4 | 67.8 | 56.7 | 53.8 |
| 8 | 8 | 5.1% | 5.2% | 7.56 | 39.6 | 72.5 | 77.8 | 68.9 | 57.0 | 53.5 |
| 16 | 4 | 8.1% | 5.2% | 7.52 | 39.0 | 72.6 | 77.9 | 68.5 | 56.9 | 53.5 |
| 32 | 2 | 11.4% | 5.3% | 7.54 | 37.4 | 69.1 | 77.7 | 68.7 | 57.0 | 53.4 |
| 60 | 1 | 15.2% | 5.5% | 7.53 | 37.2 | 70.0 | 77.1 | 68.6 | 56.8 | 53.3 |

### C.3. Ablation on Sketch

*Table 8.* Ablation study on the low-rank approximation method in 2-bit $\text{TILEQ}_S$ on Qwen1.5-MoE-A2.7B. We fix the tile rank to 32, which is the default setting used in our main evaluations. $\text{Time}_{tot}$ and $\text{Time}_{low}$ denote the total Tiling time and the time spent on low-rank approximation, respectively. R1-Sketch achieves comparable accuracy to SVD while significantly reducing the low-rank approximation overhead.

| Method | $\text{Time}_{tot}$ | $\text{Time}_{low}$ | Wiki | AC | AE | PQ | WI | MU | HS |
|---|---|---|---|---|---|---|---|---|---|
| SVD | 24.8m | 18.8m | 7.53 | 37.8 | 70.5 | 77.8 | 69.0 | 57.6 | 54.0 |
| R1-Sketch | 8.3m | 1.4m | 7.54 | 37.5 | 70.8 | 77.6 | 68.9 | 57.5 | 54.1 |

Table 8 reports the ablation results of the low-rank approximation method in $\text{TILEQ}_S$. Compared with the standard SVD-based low-rank approximation, R1-Sketch reduces the low-rank approximation time from 18.8 minutes to 1.4 minutes,

leading to a reduction of the total Tiling time from 24.8 minutes to 8.3 minutes. Meanwhile, the downstream performance remains nearly unchanged across WikiText perplexity and zero-shot reasoning benchmarks. For example, R1-Sketch obtains a WikiText perplexity of 7.54, which is very close to the 7.53 achieved by SVD, and shows comparable accuracy on AC, AE, PQ, WI, MU, and HS. These results indicate that R1-Sketch can substantially reduce the overhead of low-rank compensation construction while preserving the effectiveness of TILEQ.

## D. Appendix Evaluation Results

### D.1. Inference on other GPU Architecture

We conducted additional MoE-block inference latency evaluations on RTX 5090 (Blackwell architecture) and H800 (Hopper architecture). The results are presented in Figures 6 and 7. The findings are consistent with those observed on the A800: conventional expert-parallel approaches exhibit high latency under sparse MoE configurations, whereas the TileQ scheme—by fusing low-rank operations—achieves consistently low latency across MoE models with varying sparsity levels, delivering higher speedups as model sparsity increases.

Furthermore, TileQ's fused computation strategy remains highly effective under the 1D-Sharing paradigm, which can be interpreted as a special case of 2D tiling with row $= 1$. Compared to the tradidional 1D low-rank computation employed in Milo, TileQ achieves a $3\times$ to $20\times$ speedup.

**Reason**: The significant acceleration achieved by TileQ primarily stems from two key factors:

*Prefill Stage*: During prefill, computation is predominantly compute-bound. TileQ improves computational efficiency by fusing multiple fine-grained GEMM operations into a single large-scale matrix multiplication. This substantially increases the arithmetic intensity, enabling better utilization of tensor-core hardware and achieving over 60% of the peak compute throughput for the current batch size.

*Decode Stage*: In contrast, decoding is memory-bound. Here, TileQ eliminates repeated reads of the input matrix caused by the Top-$K$ routing mechanism. Instead of performing weighted aggregation over $\mathbb{R}^i$ tokens, it operates on a compressed low-rank intermediate representation of $\mathbb{R}^r$, thereby reducing memory access by more than $20\times$. Additionally, it reduces the number of kernel calling from $K$—to constant (refer to Figure 5). Together, these optimizations enable TileQ to maintain exceptional computational efficiency even in the decode phase.

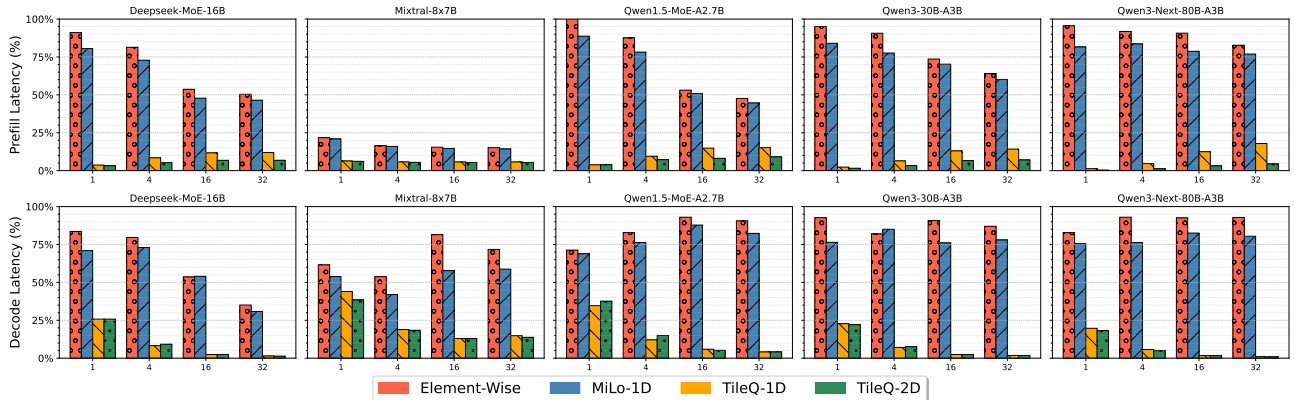

*Figure 6.* Inference latency in MoE MLP-block on H800

### D.2. Compare with other MoE methods

We noticed that several works on low-bit PTQ for MoE models have been published or accepted around the same time as our submission, including MOEQUANT (Chen et al., 2025c), MILO (Huang et al., 2025), MXMOE (Duanmu et al., 2025), and LOPRO (Gu et al., 2026).

For the remaining three methods— MILO, and MXMOE—we were unable to conduct complete and fair comparisons due to multiple practical constraints:

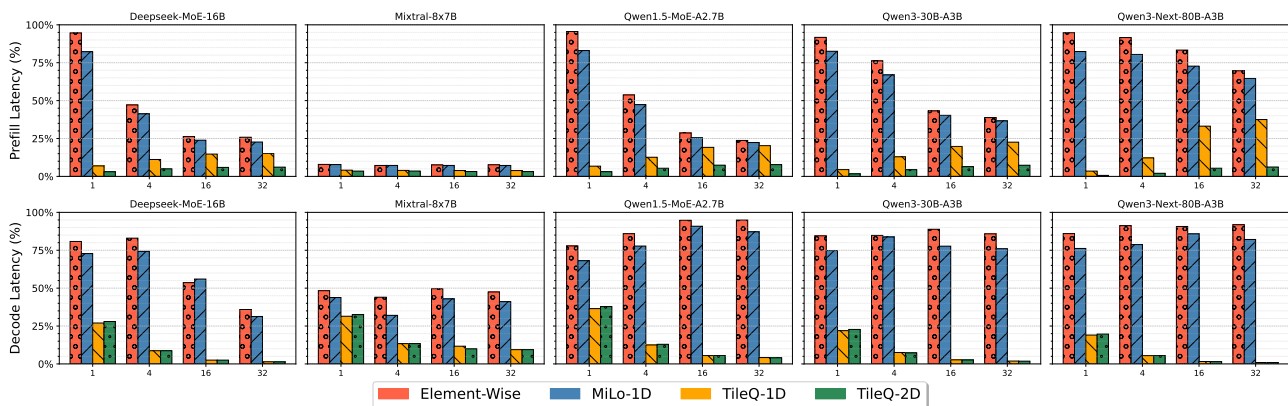

*Figure 7.* Inference latency in MoE MLP-block on 5090

- MILO and MXMOE lack sufficient implementation details (e.g., sequence length, average bit) and do not provide support for the newer model families evaluated in our work;

- MXMOE further relies on custom operators or hardware-specific primitives that are incompatible with our experimental infrastructure.

Nevertheless, to situate our approach within the broader research landscape, we include in this appendix a brief overview of these concurrent methods and perform limited case-wise comparisons wherever possible. Specifically, we extract reported results from their original papers under settings that align with ours—such as identical base models (e.g., Qwen1.5-MoE), quantization bitwidths (e.g., 2,3bit), and evaluation benchmarks (e.g., WikiText)—and compare them with our own results obtained under the same conditions.

*Table 9.* Compare with MXMOE.

| Model | Method | Avg.bit↓ | AC↑ | AE↑ | PQ↑ | WI↑ | PPL↓ |
|---|---|---|---|---|---|---|---|
| Qwen1.5-MoE | MXMOE | 2.25 | 31.7 | 53.3 | 71.3 | 61.2 | 8.79 |
| | TILEQ$_s$ | **2.16** | **39.6** | **67.3** | **77.6** | **68.9** | **7.56** |
| | MXMOE | 3.25 | **43.8** | 66.0 | 79.1 | 68.0 | 7.02 |
| | TILEQ$_s$ | **3.16** | 43.1 | **69.1** | **79.9** | **69.4** | **6.94** |
| Mixtral-8x7B | MXMOE | 2.25 | 49.0 | 72.8 | 76.3 | 68.9 | 5.63 |
| | TILEQ$_s$ | **2.16** | **58.8** | **81.3** | **81.5** | **75.1** | **4.98** |
| | MXMOE | 3.25 | 64.3 | 84.2 | **84.2** | 75.9 | 4.15 |
| | TILEQ$_s$ | **3.16** | **65.6** | **85.5** | 83.8 | **76.4** | **4.10** |

**Compare with MxMoE.** The comparison between TILEQ in scalar quantization with MXMOE are shown in Table 9. We note that the downstream tasks metric reported for MXMOE corresponds to the *acc_norm* score used in their original paper, rather than *acc* in lm-evaluation-harness (Gao et al., 2024). To ensure a fair comparison, our TILEQ$_s$ results in this table are evaluated using the same *acc_norm* protocol, which is different form Table 1.

The results demonstrate that at the 2-bit setting, TILEQ$_s$ consistently outperforms MXMOE across all metrics—achieving notably higher scores in all tasks (AC, PQ, PPL...). For instance, on Mixtral-8x7B, TILEQ$_s$ reduces perplexity from 5.63 to 4.98 and improves AE by 8.5 points. At the 3-bit level, TILEQ$_s$ is still maintaining advantages in most categories (e.g., +0.3 in WI and -0.05 in PPL on Mixtral).

**Compare with MiLo.** Then we make a comparison against MILO (Huang et al., 2025), which only reports results at approximately 3-bit level quantization. Notably, the "HS" metric in their work is also based on *acc_norm* and we adopt the same evaluation protocol for consistency. As shown in Table 10, TILEQ consistently outperforms MILO across all evaluated metrics and models. On Deepseek-MoE, TILEQ improves PIQA by 1.3, HellaSwag by 3.9, and MMLU by 2.4 percent

while simultaneously reducing perplexity from 6.42 to 6.24. The advantage becomes even more pronounced on the larger Mixtral-8x7B model with fewer memory costs (avg.bit: 3.16 vs. 3.40).

*Table 10.* Compare with MiLo.

| Model | Method | Avg.bit | PQ | HS | MU | PPL |
|---|---|---|---|---|---|---|
| Deepseek-MoE | MiLo | 3.4 | 78.9 | 74.6 | 41.9 | 6.42 |
| | TileQ$_s$ | 3.16 | 80.2 | 78.5 | 44.3 | 6.24 |
| Mixtral-8x7B | MiLo | 3.4 | 81.3 | 82.2 | 67.1 | 4.03 |
| | TileQ$_s$ | 3.16 | 83.8 | 86.0 | 69.5 | 4.12 |

**Summary.** Across all MoE-specific PTQ methods—LoPro, MxMoE, MiLo, and MoEQuant— TileQ demonstrates consistently stronger performance under comparable or even more aggressive quantization budgets. These comparisons collectively prove that TileQ offers a more robust and effective approach to low-bit PTQ for MoE models.

# E. Limitations and Future Work

**Computational Bottleneck in Quantization.** The end-to-end runtime of TileQ is dominated by the post-decomposition quantization step (e.g., GPTQ or GPTVQ), particularly for sparse MoE models with numerous small experts (e.g., Qwen3-30B-A3B). Although the 2D tiling and low-rank decomposition stages are computationally lightweight, the quantization backend remains a scalability bottleneck. A key direction for future work is the co-design of quantization algorithms tailored specifically for low-rank–structured MoE weights, potentially integrating quantization-aware low-rank optimization to reduce calibration overhead.

**Integration with Complementary Compression Techniques.** This work focuses exclusively on low-rank quantization. An important avenue for future exploration is the synergistic combination of TileQ with other compression paradigms, such as structured pruning (to remove redundant experts or tokens) or knowledge distillation. Such hybrid approaches could yield further reductions in model size and latency without compromising task performance.

