# OpenReview forum: "TileQ: Efficient Low-Rank Quantization of Mixture-of-Experts with 2D Tiling"
_ICML.cc/2026/Conference — ICML 2026 regular_

### Official Review · Reviewer_WxJX · 2026-03-09

**Soundness:** 3
**Presentation:** 3
**Significance:** 3
**Originality:** 3
**Overall Recommendation:** 4
**Confidence:** 4

**Summary:**

The paper introduces TileQ, a training-free post-training quantization (PTQ) method specifically designed to compress MoE models. To overcome the high memory overhead of per-expert low-rank decomposition and the high latency of 1D-shared approaches, TileQ clusters experts based on their singular vectors and arranges them in a 2D grid. This allows experts to share both input and output low-rank factors. Furthermore, it proposes LoTileMoE, a fused inference operator that converts irregular, sparse expert dispatch into dense matrix multiplications. Experiments across several MoE architectures demonstrate that TileQ preserves SOTA accuracy at 2-bit/3-bit settings while reducing extra memory usage by up to 10x and cutting latency overhead to around 5%.

**Compliance With Llm Reviewing Policy:**

Affirmed.

**Final Justification:**

This paper introduces a highly novel and technically sound algorithm-system co-design (TileQ and LoTileMoE) for MoE model compression. The authors' rebuttal successfully addressed all my initial concerns, providing convincing evidence on K-Means clustering stability, practical mitigations for quantization time, and favorable performance comparisons against recent baselines like KBVQ-MoE. Given the strong empirical results and the comprehensive rebuttal, my concerns are fully resolved, and I confidently maintain my positive recommendation.

**Key Questions For Authors:**

1. The baselines for comparison might not be recent enough; could you compare against more current MoE quantization methods?
2. Regarding the severe quantization time bottleneck for sparse MoE models, are there any potential mitigations or alternative backend solvers that could be integrated into TileQ to make it scale better for models with hundreds of experts?
3. How sensitive is the final quantization accuracy to the initial K-Means clustering? Have you observed high variance in perplexity across different clustering runs, and how does the method perform if the experts' singular vectors resist clean clustering?

**Limitations:**

yes

**Strengths And Weaknesses:**

**Strengths:**
* Moving from 1D-sharing to a 2D-tiling layout by clustering activation-aware singular vectors is a highly novel and principled approach to maximizing parameter sharing across experts.
* The algorithm-system co-design is exceptional. The LoTileMoE inference technique smartly resolves the critical memory-bound bottlenecks and fragmented kernel launches in low-rank MoE quantization, turning them into hardware-friendly dense GEMMs.
* The empirical validation is robust, demonstrating strong scalability and effectiveness across diverse MoE architectures (from dense to highly sparse) and across multiple GPU architectures (A800, H800, 5090).

**Weaknesses:**
* The main text lacks direct comparisons with recent, MoE-specific quantization methods (e.g., MILO, MXMOE, MoEQuant). While the authors reasonably explain the practical difficulties of this and include an appendix discussion, the main paper currently reads as if it is only competing against generic dense PTQ baselines.
* The reliance on K-Means clustering for expert assignment raises questions about stability. It is not clear how sensitive the final model accuracy is to the clustering initialization or to scenarios where expert features do not naturally form tight clusters.

---

> ### Author Rebuttal · Authors · 2026-03-27
>
> > **Response to ` Weakness 1: The main text…`**
>
> Since MoE specific quantization is a relatively recent development, there are no baselines that comprehensively cover all models. In our main experiments, we have compared our method with MoEQuant; Other methods were not included in main evaluations due to reproducibility issues stemming from a lack of open-source availability. Furthermore, KBVQ-MoE (an MoE vector quantization method), which was accepted to ICLR '26 and is closely related to our work, also only compared against MoEQuant as the MoE baseline and a few dense baselines like GPTVQ.
>
> ---
>
> > **Response to ` Weakness 2: The main text…` and ` Question 3: How sensitive is…`**
>
> ####  **Clustering sensitivity**
> We thank the reviewer for raising this important point regarding the stability of K-Means. Across 5 different runs, the standard deviation of PPL was less than 0.05 on Qwen3-30B-A3B model, indicating that our method is highly stable and not sensitive to initialization.
>
>
> #### **Greedy Tiling Stability**
> Furthermore, we have demonstrated the effectiveness of the clustering greedy algorithm through both theoretical error analysis and experiments; The greedy algorithm employed by TileQ is sufficient to achieve favorable accuracy. The result is not sensitive to a small number of tile conflicts, and the PPL variance caused by clustering is minimal, allowing the algorithm to perform well across different models. Furthermore, we did not observe extreme clustering conflicts in the four models tested in our experiments. We also posit that, given the characteristics of MoE models where existing research indicates experts always exhibit similarity, the TileQ algorithm will consistently remain effective. For specific details, please refer to `Reviewer JY8L - Response to Weakness 1 and Question 1`.
>
> ---
>
> > **Response to ` Question 1: The baselines for…`**
>
> In our experiments, we have already compared our method against the newest baselines: MoEQuant (ICML '25), MxMoE (ICML '25), and MiLo (MLSys '25). At the time of our submission deadline, these were the most competitive and state-of-the-art MoE quantization algorithms. As for now, The most recent baseline is KBVQ-MoE (ICLR '26), which had not yet been formally accepted at the ICML26 submission time; Additionally, the table below presents a comparison in Wikitext2-PPL with KBVQ: TileQ achieves significantly higher accuracy at 2-bit, while maintaining comparable accuracy at 3-bit, demonstrating that our method is competitive in SOTA PTQ works.
> |Bit|Method|Q1.5-A2.7|Q3-30B|Mix-8x7b|
> |-|-|-|-|-|
> |2|TileQ|8.12|11.12|4.78|
> |2|KBVQ|9.61|11.87|5.39|
> |3|TileQ|7.76|9.29|4.1|
> |3|KBVQ|7.74|9.26|4.07|
>
>
> ---
>
> > **Response to `Question 2: Regarding the severe…`**
>
> In TileQ, Tiling and low-rank operations account for only a small fraction of the total quantization time (refer to Table 4). While GPTQ execution is the current bottleneck, this process is highly parallelizable. We can distribute the quantization of different experts across multiple GPUs. For a model with hundreds of experts, this multi-GPU parallel strategy can linearly reduce the wall-clock time. Given that the quantized model will be used for extensive inference, the one-time overhead is justified by the significant and sustained inference acceleration (via dense GEMMs) achieved by LoTileMoE.

---

> > ### Author Rebuttal · Reviewer_WxJX · 2026-04-03
> >
> > Thanks for your detailed replies, which address most of my concerns. I will maintain the positive score.

---

> > > ### Author Response · Authors · 2026-04-07
> > >
> > > Thank you for your feedback. We are glad that our detailed replies have addressed most of your concerns, and we sincerely appreciate you maintaining the positive score.
> > >
> > > Best regards,
> > >
> > > The Authors

---

### Official Review · Reviewer_ExXk · 2026-03-12

**Soundness:** 3
**Presentation:** 3
**Significance:** 3
**Originality:** 3
**Overall Recommendation:** 4
**Confidence:** 5

**Summary:**

This paper proposes TileQ, a post-training quantization (PTQ) method for Mixture-of-Experts models that uses 2D-tiling to share low-rank factors across both input and output dimensions of expert weights. The key insight is organizing experts into an M×N grid where row experts share input-side factors (U matrices) and column experts share output-side factors (V matrices), with a single shared singular value matrix (Σ) for all experts in the tile. The paper also introduces a fused inference kernel that consolidates multiple expert computations into dense matrix operations. Experiments demonstrate substantial memory savings (up to 10× reduction in adapter overhead) and improved inference speed (latency down to 5% of unfused baseline) while having competitive accuracy on language modeling benchmarks.

**Compliance With Llm Reviewing Policy:**

Affirmed.

**Final Justification:**

Thank you for the detailed responses. The conflict statistics and theoretical analysis address my concerns about the greedy placement algorithm. The clustering stability results and the clarification that misses in single dimensions gracefully degrade to 1D sharing are valid.

The memory overhead analysis showing that intermediate buffers are comparable to (10×32)/512 of activation memory is helpful. The 4-bit comparison against GPTQ/RTN and the decode latency table also provide useful context.

I appreciate the commitment to enlarge Figures 1 and 2 in the revision.

Two minor points remain:

1. The response on Gaussian sketch vs SVD refers to LoPRo without providing direct ablation data specific to TileQ. While I understand this is an established technique, a brief accuracy comparison would strengthen the paper.

2. The rank=1 results are informative. Consider adding a brief discussion in the main paper about the accuracy-latency tradeoff at different ranks to guide practitioners.

Overall, the rebuttal has addressed my main concerns. I maintain my rating with increased confidence. The paper makes good contributions to MoE compression, particularly in system efficiency, and the experimental validation is thorough.

**Key Questions For Authors:**

1. Can you provide statistics on tile placement conflicts during the biclustering process? Specifically, how frequently do experts fail to occupy their optimal tile position, and what is the distribution of displacement distances? This would help readers understand the practical impact of the greedy placement strategy.

2. Please include comparisons with pure weight-only quantization baselines (e.g., RTN, GPTQ at 4-bit) that do not use adapters. How much accuracy does TileQ gain over these simpler methods, and what is the corresponding inference overhead? This would clarify the value proposition of the adapter-based approach.

3. What is the peak memory usage during inference, including intermediate buffers? How does this scale with batch size and number of active experts? A detailed memory profile would help practitioners assess deployment feasibility.

4. How does TileQ interact with existing LoRA adapters for task-specific fine-tuning? Can the method be applied to models that already have LoRA adapters, or used in conjunction with post-quantization adapter-based fine-tuning methods like QA-LoRA?

5. The paper uses Gaussian sketch for approximating W_big decomposition. Have you compared this against exact SVD in terms of both accuracy and computational cost? Under what conditions does the approximation introduce noticeable degradation?

**Limitations:**

Yes in appendix.

**Strengths And Weaknesses:**

Strengths:

The 2D-tiling approach is a well-motivated extension of existing low-rank quantization methods. By sharing factors bidirectionally, the method achieves better compression ratios than prior 1D approaches while maintaining model quality. The biclustering algorithm for organizing experts into tiles is principled, using alternating optimization to minimize reconstruction error.

The fused inference kernel provides substantial practical benefits. By transforming irregular expert dispatch into structured dense operations, the method improves hardware utilization. The paper provides implementation details and thorough latency analysis across different model sizes.

The experimental evaluation covers multiple model families (Mixtral, DeepSeekMoE, Qwen) and sizes. The ablation studies show the value of 2D-tiling over 1D alternatives, and the comparison with various quantization methods is informative.

Weaknesses:

The greedy placement algorithm for handling tile conflicts is a significant limitation that deserves more analysis. When multiple experts compete for the same tile position, the paper selects "nearby" positions, but provides little insight into how often this occurs and its impact on final accuracy. A frequency analysis of placement conflicts across different model architectures is necessary.

The comparison with pure quantization baselines is incomplete. While the paper compares against other low-rank methods, it lacks comparison with standard weight-only quantization methods that don't use adapters. This makes it difficult to assess the true cost-benefit tradeoff of the adapter-based approach, especially for deployment scenarios where minimizing inference complexity is critical.

The memory overhead analysis focuses primarily on adapter parameter counts but provides limited discussion of runtime memory costs. The intermediate buffers (S matrix in Algorithm 1) and the 2D accumulation structure require additional memory during inference. While these may be modest compared to activation memory, a detailed analysis would be valuable for practitioners.

The paper uses Gaussian sketch approximation instead of exact SVD for the low-rank decomposition of W_big, but does not discuss the accuracy-speed tradeoff of this choice or provide ablations comparing the two approaches.

The method's behavior under extreme compression (very small rank r) is not thoroughly explored. While the paper shows results for r=2,4,8, understanding degradation patterns at r=1 or the relationship between optimal r and tile size would provide useful insights.

Figures 1 & 2 are too small to read on printout!

The paper should explain why sharing \Sigma across all experts doesn't hurt flexibility (addressed in Section 3.2 but could be more prominent).

---

> ### Author Rebuttal · Authors · 2026-03-27
>
> > **Response to ` Weakness 1: The greedy placement…` and ` Questions 1: Can you provide…`**
>
> We have conducted statistics on tile conflicts across different models and analyzed the frequency of expert conflicts and the impact of placement on final accuracy. We conclude that although a small number of conflicts exist, the greedy tile placement algorithm is effective and achieves good performance across various models. For specific details, please refer to `Reviewer JY8L - Response to Weakness 1 and Question 1`.
>
> ---
>
> > **Response to ` Weakness 2: The comparison with…`**
>
> For accuracy, we already included standard pure W-only PTQ baselines like GPTVQ, MoEQuant, and MxMoE… For efficiency, the quantized part in TileQ is as same as most quantization formmat, which can directly leverage quantized operators in frameworks (e.g. vLLM). The additional latency introduced by TileQ is solely attributable to the low-rank adapter computations and is easy to calculate. As demonstrated in Figures 3, 5, and 7, this overhead constitutes is not a bottleneck.
>
> ---
>
> > **Response to ` Weakness 3: The memory overhead…` and `Question 3: What is the peak…`**
>
> This is a valuable point. We can analyze the memory overhead of TileQ during inference directly based on Figure 2, which faithfully reflects the logic of our implementation.
>
> Ignore the index and the small matrices such as U and V, after optimization, TileQ avoids materializing large matrices like $G \times X_{tile}$ during runtime. Instead, it employs efficient Scatter/Gather operations on smaller matrices (as shown in Figure 2c). These correspond to the `selected projs`, `weighted projs`, and `expanded tgt` tensors in the Figure 5 code with dimensions $(batchsize \times seqlen, topk \times rank)$.
>
> In activation $X$, which have dimensions $(batchsize \times seqlen, infeatures)$. Since the fist dimension is same, we only compare second dimensions:
> -   **TileQ:** $topk \times rank$. Typically, $topk< 10$, and we have demonstrated that a $rank$ of 32 is sufficient.
> -   **X** $in features$. In our experimental models, the minimum size is 512.
>
> Consequently, the additional memory overhead is comparable to the activation $(10 \times 32) / 512$. Furthermore, this memory does not accumulate across layers; Therefore, the additional memory footprint of TileQ during inference is negligible. We will include this detailed analysis in the Appendix to assist readers.
>
> ---
>
> > **Response to ` Weakness 4: The paper uses…` and `Question 5: The paper uses…`**
>
> We thank the reviewer for the opportunity to clarify that our focus is on the novel 2D-Tiling application, to prioritize originality, we refrained from excessive details and experiments on Sketch; Complete ablation studies on its accuracy and speed towards normal SVD are given in LoPRo. In summary, this method is entirely adequate for PTQ and will not cause degradation in quantization.
>
> ---
>
> > **Response to ` Weakness 5: The method's behavior…`**
>
> We evaluated the PPL and inference latency of Qwen1.5-MoE-A2.7B at rank=1 across different Tile, as shown in the table below.
> |M|N|Latency-P|Latency-D|PPL-Wiki|
> |-|-|-|-|-|
> |1|60|6.1|3.9|9.66|
> |2|30|4.7|3.2|9.64|
> |4|15|3.6|2.5|9.58|
> |8|8|3.2|2.1|9.62|
>
> As the rank decreases, the latency reduced marginal but the accuracy drops substantially. This indicates that in the extreme low-rank scenario where r=1, the matrix's primary features cannot be fully catch, leading to higher loss. In summary, when the rank is not particularly small (>=16), approximates square Tiling achieves the best overall performance in terms of accuracy, compression ratio, and inference latency (given in Appendix C.1 and C.2).
>
> ---
>
> > **Response to ` Weakness 6: Figures 1 & 2…`**
>
> We appreciate your suggestion. We will increase their size in the revised version.
>
> ---
>
> > **Response to ` Weakness 7: The paper should explain…`**
>
> The Sigma can be computed by a fast element-wise row multiplication (Fig. 2-B.1) in a small low-rank matrix. This operation is highly efficient and has negligible impact on the overall algorithm efficiency. We will clarify this in the revised manuscript.
>
> ---
>
> > **Response to ` Question 2: Please include comparisons…`**
>
> The comparison with GPTQ and RTN at 4-bit is presented in the table below.
>
> |Method|PPL|Latency-P|Latency-D|
> |-|-|-|-|
> |RTN|12.4|-|-|
> |GPTQ|8.44|-|-|
> |TileQ|8.15|2.40%|4.40%|
>
> TileQ still achieves improved accuracy and low latency compared to these two baseline methods at 4-bit. Furthermore, consistent with other studies (e.g., LQER, MILO, KBVQ), low-rank PTQ exhibits more significant advantages in low-bit regimes.
>
> ---
>
> > **Response to ` Question 4: How does TileQ interact…`**
>
> We appreciate your insightful and forward-looking question. Normal low-rank PTQ can be combined with LoRA adapters for fine-tuning, as studied in Caldera and LoPRo. However, the integration of 2D-Tiling with fine-tuning requires further exploration and can be a direction for future research.

---

> > ### Author Rebuttal · Reviewer_ExXk · 2026-04-02
> >
> > see my final justifications

---

> > > ### Author Response · Authors · 2026-04-03
> > >
> > > We are more than willing to address any further questions you may have. However, we are currently unable to see the "final justifications" you mentioned; it appears that the response is not visible to the authors.
> > >
> > > Could you please kindly repost your comments or check the access settings?
> > >
> > > Best regards,
> > >
> > > The Authors
> > >
> > > ---
> > > ---
> > > Dear Reviewer,
> > >
> > > We are now able to view your comments in the Final Justification. Thank you for your thorough review and for acknowledging our responses in the rebuttal. Regarding your follow-up questions, we made the following response
> > >
> > > - We confirm that we will include additional experiments regarding the use of Sketch and SVD techniques in TileQ in the revised version to further strengthen the paper's validity.
> > >
> > > - We will include the discussion on the accuracy-latency trade-off for different ranks (such as the rank=1 series) from our rebuttal into the paper's Appendix. We believe this will better assist practitioners in applying our method.
> > >
> > > Best regards,
> > >
> > > The Authors

---

### Official Review · Reviewer_wyLK · 2026-03-13

**Soundness:** 3
**Presentation:** 4
**Significance:** 3
**Originality:** 3
**Overall Recommendation:** 4
**Confidence:** 4

**Summary:**

This paper presents TileQ, a post-training quantization method for MoE models. The main idea is to place experts on a 2D grid and share low-rank factors along both grid dimensions. This reduces low-rank memory overhead and also enables a more efficient fused inference path. The method is evaluated on five MoE models under 2-bit and 3-bit settings. The main benefit is lower low-rank overhead with modest latency cost.

**Compliance With Llm Reviewing Policy:**

Affirmed.

**Final Justification:**

All of my concerns have been solved, and based on the contribution of this paper, I prefer to keep my positive score.

**Key Questions For Authors:**

1. At a matched extra-bits budget, does TileQ outperform MILO in accuracy, or mainly in memory and runtime?
2. How far are final expert placements from their ideal bicluster positions, especially for very large MoE models?
3. Can the paper provide a clearer decode latency summary across models?

**Limitations:**

1. The evaluation does not include end-to-end generation quality.
2. The paper’s strongest contribution is systems and memory efficiency, not clearly better quantization quality.

**Strengths And Weaknesses:**

Strengths:
The paper identifies a real bottleneck in low-rank MoE quantization and addresses it with a 2D tiling design. By sharing low-rank factors across both row and column groups of experts, it significantly reduces the memory overhead of low-rank compensation, which becomes especially important for large-expert MoE models. The paper also adds a practical fused inference implementation to reduce irregular expert-wise computation, and validates the method on a broad set of models, bit-widths, and hardware platforms.

Weaknesses:
1. The paper overstates the novelty on the quantization side. The main gain is memory efficiency, not a clear improvement in quantization accuracy. TileQ still depends on existing components such as GPTQ or GPTVQ and LoPRo-style rotation.
2. Greedy placement has no quality guarantee. The placement algorithm solves Eq. 9 via a Chebyshev concentric search heuristic. There is no approximation ratio, no analysis of how far the solution is from optimal, and no discussion of degradation when K >> M×N or how much it degrades at larger expert counts.

---

> ### Author Rebuttal · Authors · 2026-03-27
>
> > **Response to `Weakness 1: The paper overstates…` and `Limitation 2: The paper’s strongest`**
>
> We appreciate the reviewer's insightful observation. We agree that the primary contribution of TileQ lies in system efficiency (reducing memory overhead and irregular computation) rather than introducing a fundamentally new quantization metric that outperforms GPTQ/GPTVQ in pure accuracy.
> TileQ is designed as a system-aware quantization framework for MoE models. Our goal is to demonstrate that by introducing 2D tiling and spatial allocation, we can achieve competitive accuracy (leveraging existing techniques like GPTQ) while drastically improving memory and runtime efficiency. We have revised the manuscript to better emphasize this system-centric contribution and clarify that the quantization components are built upon established foundations to ensure robustness.
>
> ---
>
>
> > **Response to `Weakness 2: Greedy placement has…` and ` Questions 2: How far are…`**
>
> We address the concerns regarding the theoretical guarantees of our greedy placement algorithm and the performance degradation at scale through both theoretical analysis and empirical validation.
>
> #### **1. Constraint on Expert Count ($K$) and Grid Size ($M \times N$)**
> We clarify that in our framework, the number of experts $K$ does not exceed the grid capacity $M \times N$. The dimensions $M$ and $N$ are explicitly determined based on $K$ to ensure sufficient capacity. Our experiments indicate that performance is optimal when $M \times N > K$ and the grid shape approximates a square matrix. Furthermore, we observe that relative execution efficiency improves as the number of experts increases, demonstrating the scalability of our approach.
>
> #### **2. Analysis of Conflicts and Optimality**
> We have conducted extensive statistical analysis on tile conflicts across various models. While we acknowledge that a greedy approach does not offer a strict approximation ratio, our results show that the frequency of expert conflicts is low and has a negligible impact on final accuracy.
>
> In TileQ, when a placement conflict occurs, we employ a heuristic that prioritizes selecting sub-blocks adjacent in either rows or columns. This strategy ensures that simultaneous misses in both dimensions are rare. In the event of a miss in only one dimension (row or column), the algorithm degrades gracefully to a 1D sharing mode along that specific dimension (similar to Milo). These design choices collectively ensure the robustness and accuracy of TileQ, effectively compensating for the lack of a theoretical approximation bound.
>
> #### **3. Full results and theoretical analysis**
> For specific details regarding the conflict statistics and theoretical analysis, please refer to our response to `Reviewer JY8L - Response to Weakness 1 and Question 1`.
>
> ---
>
> > **Response to ` Questions 1: At a matched extra-bits…`**
>
> As shown in Table 9, when comparing TileQ with MILO at matched extra-bits. TileQ achieves superior accuracy with lower quantization memory overhead, but MILO holds an advantage in terms of runtime. This is primarily attributed to the poor efficiency of the GPTQ algorithm when running on MoE experts (whereas 2D Tiling accounts for less than 10% of the time). Furthermore, we believe that as long as the runtime is not excessively high, it is acceptable for a quantization algorithm (e.g., single-GPU quantization of Mixtral-8x7b is completed within 3 hours). Moreover, this issue can be significantly mitigated through multi-GPU parallelism (i.e., quantizing the experts of a single layer simultaneously across multiple GPUs).
>
> ---
>
> > **Response to ` Questions 2: Can the paper provide…`**
>
> We have presented the prefill and decode inference latencies for five models across three GPU architectures in Figures 3, 6, and 7. We summarize the MLP inference latency of TileQ during the decode phase on the A800 in the table below:
>
> |Config|Batch|Latency-TileQ|
> |-|-|-|
> |Q1.5-A2.7B|1|35.3%|
> ||4|13.1%|
> ||16|5.3%|
> ||32|3.9%|
> |Q3-A3B|1|20.4%|
> ||4|7.0%|
> ||16|2.6%|
> ||32|1.8%|
> |Q3-Next |1|15.8%|
> ||4|5.7%|
> ||16|1.6%|
> ||32|0.9%|
> |Mixt-8x7B|1|38.6%|
> ||4|17.6%|
> ||16|12.9%|
> ||32|12.8%|
> |DS-MOE|1|25.1%|
> ||4|8.5%|
> ||16|2.5%|
> ||32|1.4%|
>
>
> ---
>
> > **Response to ` Limitation 1: The evaluation…`**
>
> We have included end-to-end generation quality evaluations in Tables 1, 6, 8, and 9, which report perplexity and task accuracy (e.g., on MMLU) for various models and bit-widths. These results demonstrate that TileQ maintains end-to-end model performance.

---

> > ### Author Rebuttal · Reviewer_wyLK · 2026-04-03
> >
> > Most of my concerns have been solved, and based on the contribution of this paper, I prefer to keep my positive score.

---

> > > ### Author Response · Authors · 2026-04-07
> > >
> > > Thank you for your feedback and for confirming that most of your concerns have been resolved. We appreciate your continued positive evaluation of our work.
> > >
> > > Best regards,
> > >
> > > The Authors

---

### Official Review · Reviewer_JY8L · 2026-03-13

**Soundness:** 3
**Presentation:** 1
**Significance:** 3
**Originality:** 3
**Overall Recommendation:** 4
**Confidence:** 3

**Summary:**

The paper proposes TileQ, a fine-tuning-free post-training quantization method for Mixture-of-Experts (MoE) models that utilizes a 2D-tiling structured low-rank quantization approach. By sharing low-rank factors across both the input and output dimensions of MoE, TileQ aims to mitigate the high memory overhead typically associated with low-rank MoE quantization. Additionally, the authors introduce a fused inference technique, LoTileMoE, which combines multiple sparse expert computations into regular dense tensor operations to accelerate both the prefill and decode phases. The method is evaluated on several MoE architectures, demonstrating strong accuracy at extremely low bit-widths (2-bit and 3-bit) while reducing memory overhead and inference latency compared to per-expert and 1D-shared baselines

**Compliance With Llm Reviewing Policy:**

Affirmed.

**Key Questions For Authors:**

How frequently does the greedy placement algorithm fail to place an expert near its ideal cluster center? What is the empirical impact of these spatial displacements on the actual task loss compared to an optimal assignment solver?

High latency during quantization inference is somewhat expected, as native 2-bit and 3-bit operations are generally unsupported, requiring excessive amounts of quantization and dequantization operators within the kernels. What is the relative latency overhead of the 2D-tiling method in a standard 4-bit setting on modern GPUs that natively support 4-bit tensor core operations?

**Limitations:**

yes

**Strengths And Weaknesses:**

The LoTileMoE inference formulation successfully replaces irregular, sparse expert dispatch loops with highly parallelized dense GEMMs and vectorized scatter-adds, addressing a major latency bottleneck in existing MoE PTQ methods.
TileQ demonstrates impressive robustness at ultra-low precision (2-bit), outperforming existing non-finetuned baselines across diverse MoE architectures (e.g., Mixtral-8x7B, Qwen1.5-MoE) while maintaining minimal latency overhead.

To resolve spatial collisions in the 2D-Tiling step, the authors rely entirely on a greedy spatial allocation strategy based on Chebyshev distance. Because it is a greedy algorithm, early assignments may force subsequent experts far away from their ideal cluster centers. Quantifying how this sub-optimal displacement degrades the "Local Subspace Proximity" error bound they rely on theoretically is not clearly addressed.

The figures in the manuscript (particularly Figures 1 and 2) are heavily pixelated, densely packed, and require excessive zooming to read the text and discern critical details. This significantly hinders the reader's ability to understand the proposed architecture and pipeline, especially when printed.

---

> ### Author Rebuttal · Authors · 2026-03-27
>
> > **Response to `Weakness 1: To resolve spatial…` and `Question 1: How frequently does…`**
>
> #### **Frequency of Displacement**:
> By the frequency at which this suboptimal displacement method occurs in the model as you mentioned, We present the Greedy  Tile layout hit rates for the three models in the table below (The 'ideal Placement' is the Tiling Map that results in the minimum MSE error after arranging the experts of this layer, which requires a significant amount of time for calculation).
>
> |Type|Qwen30b-A3|Deepseek|Mixtral|
> |-|-|-|-|
> |Ideal Placement|0.66|0.65|0.55|
> |Row Miss|0.17|0.18|0.22|
> |Col Miss|0.13|0.11|0.15|
> |R&C Miss|0.04|0.06|0.08|
>
> Due to the intrinsic of the SVD algorithm, Tile partitioning yields 4 possible hit scenarios based on row and column alignments. Crucially, the quality of the low-rank approximation is invariant to the distance between blocks (permuting matrix only affects the arrangement of the $U$ and $V$, leaving the accuracy unchanged).
>
> In TileQ, when a placement is in conflict, we prioritize selecting sub-blocks that are adjacent in either rows or columns. This strategy ensures that cases with misses in both rows and columns occur rarely. When a miss occurs in only one dimension (row or column), the algorithm degrades to at most a 1D sharing mode along that specific dimension (like Milo). These designs collectively guarantee the accuracy of TileQ.
>
> #### **Theoretical & Empirical**:
> We substantiate the rationality of employing the greedy clustering algorithm through three key dimensions:
> - *Theoretical Bound*: As proven in Appendix A.2, the total reconstruction error is upper-bounded by the sum of independent quantization errors and clustering displacement ($E < E_{ind} + \epsilon_{cluster}$). This establishes that TileQ's error lower bound corresponds to that of an independent low-rank for each expert.
> - *Empirical Validation*: Our ablation studies in Table 3 demonstrate that replacing 2D-Tiling with independent approximation results in a negligible accuracy degradation of less than 2%. This confirms that the "suboptimal displacement" introduced by our greedy strategy falls well within the tolerance margin of our theoretical bounds, effectively preserving the "local subspace proximity" required for high-fidelity recovery.
> - *Algorithmic Efficiency*: Overall, the greedy algorithm achieves an optimal balance between high efficiency and accuracy. While global optimal solvers (e.g., the Hungarian algorithm) could theoretically minimize displacement further, pursuing a global optimum would incur $O(K^3)$ complexity, which contradicts TileQ's core objective: enabling efficient and scalable PTQ.
>
>
> ---
>
> > **Response to `Weakness 2: The figures in…`**
>
> We thank for this valuable suggestion. In the initial submission, due to strict page limits, we reduced the size of figures; we will enlarge Figures 1 and 2 to ensure text and details are legible without excessive zooming.
>
> ---
>
> > **Response to `Question 2: High latency during…`**
>
> We conducted latency tests on Linear layers using the 4-bit kernels from GPTQModel (as in Figure 3, batch size=4). The results are analyzed as follows:
>
> |Latency|Qwen30b-A3|Deepseek|Mixtral|Qwen3-Next|
> |-|-|-|-|-|
> |Prefill|3.3%|5.5%|6.1%|2.4%|
> |Decode|6.2%|5.2%|9.8%|4.1%|
>
> Since inference latency only comes from low-rank components, the execution time remains the same across different bit-widths, as they are computed in FP16.
> In prefill, while the W4 operators demonstrate slightly higher computational efficiency. In decode, W4 layers suffer from bottlenecks in weight loading, and TileQ achieves lower latency than 2-bit and 3-bit. In summary, TileQ delivers superior inference latency across 4-bit, 3-bit, and 2-bit configurations.

---

> > ### Author Rebuttal · Reviewer_JY8L · 2026-04-04
> >
> > Thank you for the rebuttal. You have adequately addressed my questions, and I will maintain my positive score.

---

> > > ### Author Response · Authors · 2026-04-07
> > >
> > > Thank you very much for your feedback and for confirming that your concerns have been adequately addressed. We appreciate you maintaining your positive evaluation of our work.
> > >
> > > Best regards,
> > >
> > > The Authors

---

### Decision · Program_Chairs · 2026-04-30

**Decision:**

Accept (regular)

**Comment:**

In their paper, the authors introduce a novel approach called TileQ, which aims to improve modeling
performance and efficiency through a carefully designed methodological innovation in post-training
quantization (PTQ) for Mixture-of-Experts (MoE) models. Reviewers generally agreed that the core
idea—specifically the use of 2D-tiling structured low-rank quantization—is interesting and original,
and that the empirical results demonstrate consistent improvements across a range of settings, including
high-bit and low-bit quantization for models like Mixtral and DeepSeek. The paper is well motivated,
and the proposed method addresses the critical problem of deployment efficiency in massive MoE
architectures. However, several weaknesses were raised during the review process, some of which
were only partially addressed in the rebuttal.

**Recommendation:** Given the novelty of the proposed idea, the solid empirical validation, and
the authors’ efforts to address reviewer feedback regarding computational overhead during the rebuttal,
I believe the strengths of the paper outweigh its weaknesses. While there is room for improvement
in terms of broader evaluation and theoretical depth, these limitations do not detract from the overall
contribution to model compression. I therefore recommend accepting the paper, and I encourage the
authors to further strengthen the theoretical analysis and expand the experimental scope in future work.